# CRISPR-Cas9 engineering of the *RAG2* locus via complete coding sequence replacement for therapeutic applications

Daniel Allen[1,5], Orli Knop[1,5], Bryan Itkowitz[1,5], Nechama Kalter [1], Michael Rosenberg[1], Ortal Iancu[1], Katia Beider[2], Yu Nee Lee[3,4], Arnon Nagler[2,3], Raz Somech[3,4] & Ayal Hendel [1]✉

*RAG2*-SCID is a primary immunodeficiency caused by mutations in *Recombination-activating gene 2 (RAG2)*, a gene intimately involved in the process of lymphocyte maturation and function. ex-vivo manipulation of a patient's own hematopoietic stem and progenitor cells (HSPCs) using CRISPR-Cas9/rAAV6 gene editing could provide a therapeutic alternative to the only current treatment, allogeneic hematopoietic stem cell transplantation (HSCT). Here we show an innovative *RAG2* correction strategy that replaces the entire endogenous coding sequence (CDS) for the purpose of preserving the critical endogenous spatiotemporal gene regulation and locus architecture. Expression of the corrective transgene leads to successful development into CD3+TCRαβ+ and CD3+TCRγδ+ T cells and promotes the establishment of highly diverse TRB and TRG repertoires in an in-vitro T-cell differentiation platform. Thus, our proof-of-concept study holds promise for safer gene therapy techniques of tightly regulated genes.

Severe combined immunodeficiency (SCID) is a group of multiple rare monogenic disorders characterized by defects in both cellular and humoral adaptive immunity. Patients are born healthy and due to being extremely susceptible to pathogens, they present with recurrent infections early in life which if left untreated can be fatal[1–3]. The *Recombination-activating genes 1* and *2* (*RAG1* and *RAG2*, respectively) are tightly linked and have convergent transcriptional orientations on chromosome 11 separated by ~12 kb. The *RAG* genes encode proteins that, when complexed together, commence the lymphoid-specific variable (V), diversity (D), and joining (J) gene [V(D)J] recombination process by catalyzing DNA double-strand breaks (DSBs) at the recombination signal sequences (RSSs) which flank the V, D, and J gene segments[4]. V(D)J recombination is a critical step in the maturation of T and B cells as it is responsible for the generation of a diverse repertoire of T- and B-cell receptors (TCR and BCR, respectively). Thus, patients

with disease-causing variants in the *RAG* genes typically present with the complete absence or significant reduction of T and B cells and the T-B-NK+ immune phenotype[2,5–9]. V(D)J recombination has three main mechanisms of regulation: 1) lineage specificity, namely BCR and TCR gene rearrangement occurs in B cells and T cells, respectively; 2) immunoglobulin heavy-chain rearrangement occurs before immunoglobulin light chain; and 3) allelic exclusion, namely once a T or B cell rearranges its receptor locus, only one functional allele will be expressed in that cell. In addition to these general mechanisms, the transcription of the *RAG1* and *RAG2* genes is regulated by numerous highly-conserved, lineage-specific, cis-acting sequences surrounding their respective CDSs which control the spatial genomic organization inside the locus[10–15]. The *RAG* genes are expressed exclusively in the $G_0$/$G_1$ stage and display a tightly linked genomic organization with regulated expression during specific phases of T and B lymphocyte

[1]Institute of Nanotechnology and Advanced Materials, The Mina and Everard Goodman Faculty of Life Sciences, Bar-Ilan University, Ramat-Gan 5290002, Israel. [2]The Division of Hematology and Bone Marrow Transplantation, Chaim Sheba Medical Center, Tel-Hashomer, Ramat Gan 5266202, Israel. [3]Sackler Faculty of Medicine, Tel Aviv University, 6997801 Tel Aviv, Israel. [4]Pediatric Department A and the Immunology Service, Jeffrey Modell Foundation Center, Edmond and Lily Safra Children's Hospital, Sheba Medical Center, Ramat Gan 5266202, Israel. [5]These authors contributed equally: Daniel Allen, Orli Knop, Bryan Itkowitz. ✉e-mail: ayal.hendel@biu.ac.il

development. More specifically, two limited waves of *RAG1* and *RAG2* expression are necessary for Ig heavy and light chain rearrangements after which their expression is promptly terminated[16]. This process is orchestrated by a plethora of transcription factors and machinery that complex together with cis-regulatory elements and promoter regions in the *RAG1/2* locus. Together, they create a chromatin hub that acts as a super-enhancer for the expression of the *RAG* genes[12,14,17,18]. This formation, as well as the chromatin structure and 3D architecture, are crucial to ensuring that the *RAG1/2* genes are only expressed during the requisite developmental window[19]. Overexpression or expression of the *RAG* genes outside of this precise window can result in genomic instability and lymphocyte malignancy through the formation of translocations and/or deletions in cancer-causing genes[11,12,19–23]. Unsuccessful termination of *RAG1* and *RAG2* expression is also associated with atypical thymus development, an aberrant lymphatic system, and immunodeficiency[24].

Currently, the only definitive curable treatment for SCID patients is allogeneic hematopoietic stem cell transplantation (HSCT) from a human leukocyte antigen (HLA)-matched donor[25]. However, finding a HLA-matched donor is rare, and the alternative treatment, haploidentical HSCT, reduces the survival rate from >80% with an HLA-matched donor to 60-70%[26,27]. Although successful HSCT promotes lymphoid lineage development resulting in a long-term patient survival rate, it is accompanied by a high risk of graft-versus-host disease[28,29].

An ideal alternative to searching for an HLA-matched donor is to genetically edit the patient's own CD34+ HSPCs ex-vivo to be subsequently returned to the patient as an autologous HSCT. CD34+ HSPCs' marked ability to reconstitute the immune system from a small number of cells, makes these cells an attractive platform for gene-therapy applications[30,31]. To that end, transgene delivery via lentiviral (LV) or gammaretroviral (γRV) vectors for ex-vivo editing of patients' CD34+ HSPCs was previously reported in gene therapy clinical trials[32–36]. However, genome-editing-based treatments using γRV of Chronic granulomatous disease (CGD)[37], Wiskott-Aldrich syndrome (WAS)[38,39], SCID-X1[40], and Adenosine deaminase (ADA)-SCID[41] resulted in the activation of proto-oncogenes leading, in some patients, to a leukemic transformation[42,43]. Although steps have been taken to improve the safety of these viral vectors, transgene integration into tumor-suppressor loci has been observed and incomplete phenotypic correction, toxicity, dysregulated hematopoiesis, and insertional mutagenesis related to the semi-random integration and constitutive expression of the transgene into the genome remain major safety concerns[44,45]. Despite these concerns, which are particularly severe for highly controlled and regulated genes such as the *RAG1/2*, LV-based gene therapy for *RAG1* is currently undergoing clinical trials[46].

For these reasons, transgene delivery via a targeted HDR-mediated genome-editing approach such as the combination of clustered regularly interspaced short palindromic repeats (CRISPR) and CRISPR-associated nuclease 9 (Cas9) and recombinant adeno-associated virus serotype 6 (rAAV6) could prove particularly beneficial for treating *RAG*-SCIDs[47–49]. Although rAAV6 provides a number of benefits over LV and γRV vectors, we have shown previously that rAAV6 vectors trigger a toxic DNA damage response (DDR) proportional to the multiplicity of infection (MOI), or the amount of virus used[50]. Thus, for a CRISPR-Cas9/rAAV6 genome-editing strategy to be therapeutically relevant, a delicate balance between maintaining high-quality HDR while reducing viral load as much as possible is required. Therefore, we tested a number of different donor constructs in order to find the optimal rAAV6 donor design for efficient HDR.

Additionally, a highly specific CRISPR-Cas9-based approach is particularly beneficial for disorders where the affected gene is tightly regulated, such as *RAG*-SCIDs[11,12,51]. CRISPR-Cas9/rAAV6 genome editing provides the potential for specific gene integration that maintains

the strict endogenous spatiotemporal gene regulation and expression of the transgene. Modulation of transgene expression levels by including mRNA stability and/or nuclear export cis-acting elements such as polyadenylation (polyA or pA) signals or woodchuck hepatitis virus posttranscriptional regulatory element (WPRE) segments has been widely reported[52]. Polyadenylation at the end of a RNA transcript affects mRNA stability, nuclear export, and translation, thus, playing a crucial role in gene expression. In-vitro studies using rAAV6 vectors have highlighted the effects of different synthetic polyA signals, such as bovine growth hormone (BGH) polyA signal sequences, in boosting transgene expression[53,54]. Similarly, cis-acting post-transcriptional regulatory elements (PREs), such as WPRE, are required for the nuclear export of intronless RNA, and the addition of such PREs has been reported to enhance transgene expression in-vitro[53,55]. Through the use of such sequences to regulate our *RAG2* transgene, we aimed to establish transgene expression that closely resembles that of the endogenous gene.

We previously reported on a proof-of-concept CRISPR-Cas9/rAAV6 *RAG2* gene correction approach[56] via insertion of the corrective transgene into the CRISPR-Cas9-induced cut site adjacent to the *RAG2* start codon. This approach successfully led to the development of T cells with diverse TCR repertoires from *RAG2*-SCID patient-derived CD34+ HSPCs. Here, we describe a CRISPR-Cas9/rAAV6-mediated genome-editing approach that replaces the entire *RAG2* CDS with a corrective transgene in CD34+ HSPCs, to maintain the endogenous spacial regulatory elements of the *RAG2* locus. We achieved this by constructing three *RAG2* CDS replacement correction donors where each maintains the 5′ endogenous promoter and regulatory elements[10,15,57–59]. However, in addition to the regulatory elements upstream to the *RAG2* gene, at least one silencer sequence has been discovered between the *RAG1* and *RAG2* genes that is counteracted by an anti-silencing element 5′ of the *RAG2* promoter[60]. Additionally, there are evolutionarily conserved genes within the *RAG1/2* locus' intronic sequences that are believed to play a role in transcriptional regulation[11,12,61,62]. Thus, to most closely mimic the endogenous expression of *RAG2*, we designed one donor to preserve the 3′ UTR and downstream region in addition to the 5′ UTR and promoter region. The two additional correction donors were engineered in an effort to increase the expression levels of the transgene by maintaining the *RAG2* 5′ UTR endogenous region while introducing synthetic 3′ UTRs (WPRE-BGHpA and BGHpA sequences).

Developing a proof-of-concept gene-correction therapy typically requires large amounts of patient samples. However, since untreated SCID patients rarely survive past infanthood, neither invasive bone marrow procedures nor drawing large volumes of peripheral blood (PB) are viable options for procuring such samples. In this work, to circumvent this challenge, we used fluorescence-activated cell sorting (FACS) enrichment of healthy donor (HD)-derived CD34+ HSPCs with engineered genotypes after multiplex HDR (knock-in/knock-out [KI-KO]) to simulate single-allelic gene-correction therapies for *RAG2*-SCID. Since *RAG2*-SCID is an autosomal recessive disorder, correction of only one mutated allele is sufficient to cure the patient. In this strategy, we mimicked monoallelic correction in SCID-patient cells by knock-in (KI) of a diverged codon-optimized *RAG2* (dco*RAG2*) cDNA cassette in place of the endogenous *RAG2* CDS in one allele (thereby preserving regulatory non-coding elements) and by knock-out (KO) of the second allele with a green fluorescent protein (GFP) gene-disrupting cassette. The dco*RAG2* cDNA produces a protein identical to wild-type (WT) RAG2, while the introduction of wobble changes leads to reduced similarity to the genomic sequence precluding the Cas9 from re-cutting the inserted sequence or from the inserted sequence serving as a homology arm causing premature cessation of HDR. Via cell sorting, we were then able to enrich for cells with the desired KI-KO-engineered genotype to model and track their progression into T-cell development. Thus, in this study, we show *RAG2*

gene correction simulation in HD-derived CD34$^+$ HSPCs by replacement of the entire endogenous CDS.

## Results

### Different HDR strategies: cut-site insertion vs. coding sequence (CDS) replacement and adjusting homology arm length

*RAG2*-SCID is caused by mutations scattered throughout the CDS of the *RAG2* gene[4]. Therefore, a universal correction technique that would suit all *RAG2*-SCID patients requires the delivery of an intact copy of the complete *RAG2* CDS. While *KI* of an intact CDS at the endogenous locus would achieve this, this strategy could interfere with the 3D chromatin architecture and critical endogenous gene regulation by moving regulatory elements further downstream from the transgene. This could potentially disrupt spatial cross-talk between functional elements upstream and downstream of the *RAG2* gene, such as promoter and/or enhancer sequences[12,14,17,18] (Supplementary Fig. 2A–D). Hence, we hypothesized that a donor DNA with a left homology arm (LHA) upstream adjacent to the cut site, and a right homology arm (RHA) distanced from the cut site, downstream of the *RAG2* stop codon would ensure preserving *RAG2* regulatory elements by replacing the entire *RAG2* CDS. To examine the feasibility of our hypothesis, we produced two rAAV6 vectors that would integrate a GFP expression cassette under the regulation of a spleen focus-forming virus (SFFV) promoter and BGHpA sequence after delivery of a chemically modified *RAG2* sgRNA/Cas9 ribonucleoprotein (RNP) complex via electroporation into CD34$^+$ HSPCs. In previous studies, we demonstrated this sgRNA's high on-target editing efficiency and accuracy[50,63,64]. The first donor[56], herein CSI_GFP-BGHpA_400×400, uses 400 bp homology arms immediately flanking the Cas9-induced cut site for donor insertion (hereafter termed a cut-site-insertion [CSI] vector), while the second donor, herein CDSR_GFP-BGHpA_400×400, uses a 400 bp LHA spanning the immediate sequence upstream to the Cas9 cut site and a 400 bp RHA spanning the immediate sequence downstream to the *RAG2* stop codon, to replace the entire *RAG2* CDS with the DNA donor (herein a CDS-replacement [CDSR] vector) (Fig. 1A, Supplementary Table 1, and Supplementary Data 1). Two days post-editing, we analyzed the frequencies of GFP$^+$ cells via flow cytometry to determine the HDR efficiencies of the different donors. Each rAAV6 donor was gated based on its respective rAAV only (RNP$^-$) sample to minimize the number of cells that are included in the positive population that are expressing the reporter in an episomal manner without HDR-mediated integration of the cassette into the genome (GFP$^{low}$ cells) (see Supplementary Note 1 and Supplementary Fig. 1 for a description of the gating strategy). We found that the HDR efficiency of the CSI_GFP-BGHpA_400×400 vector was significantly higher than that of the CDSR_GFP-BGHpA_400×400 vector (21.8% and 9.1%, respectively) (Fig. 1B and Supplementary Fig. 2E). Attempting to improve the HDR efficiency of the CDSR technique, we designed an additional two rAAV6 donors with RHAs extended from 400 bp to 800 bp and 1,600 bp spanning the immediate region downstream to the *RAG2* stop codon (herein CDSR_GFP-BGHpA_400x800 and CDSR_GFP-BGHpA_400x1600, respectively [Fig. 1A, Supplementary Table 1, and Supplementary Data 1]). While elongation to 800 bp produced significantly higher HDR efficiency than the CDSR_GFP-BGHpA_400×400 donor (14.8%), only after elongation to 1,600 bp, did we observe HDR efficiency comparable to that of the CSI_GFP-BGHpA_400x400 donor (25.2%). (Fig. 1B and Supplementary Fig. 2E). Using a uniform pair of primers and probe for droplet digital PCR (ddPCR), we confirmed that the HDR efficiencies as determined by flow cytometry were accurate and locus-specific (Fig. 1C and Supplementary Fig. 2F). To validate that our CDSR strategy is broadly applicable and not specific only to the *RAG2* locus, we designed a set of rAAV6 donors to introduce a GFP expression cassette into the *RAG1* locus (Supplementary Fig. 3A, Supplementary Table 2, and Supplementary Data 1). Similar to *RAG2*, we used a highly specific sgRNA that targeted just downstream from the

*RAG1* ATG start codon. Since *RAG1* CDS is longer than that of *RAG2*, the CDSR method here replaced 3,112 bp as opposed to only 1,541 bp at the *RAG2* locus. While we were able to achieve highly efficient HDR at the *RAG1* locus as well, we found that longer homology arms were required to do so (Supplementary Fig. 3B–E).

Interestingly, via flow cytometry, we observed a significantly higher mean fluorescence intensity (MFI) of GFP$^{high}$ cells after integration of the CDSR vectors compared to the CSI vectors (Fig. 1D, E and Supplementary Fig. 2E and Supplementary Note 1). This difference highlights that different integration strategies have unique effects and can lead to distinctive conformational changes on the genomic locus and impact subsequent transgene expression.

### Synthetic polyA sequences and/or cis-acting PREs affect transgene expression

To modulate transgene expression further, we aimed to test the impact of synthetic 3′ regulatory elements on transgene expression. Thus, we designed two CDSR vectors (herein CDSR_GFP-WPRE-BGHpA_400x1600 and CDSR_GFP-NoBGHpA_400x1600 [Fig. 2A, Supplementary Table 1, and Supplementary Data 1]) with a homology arm pattern similar to that of the CDSR_GFP-BGHpA_400x1600 donor. However, whereas the CDSR_GFP-BGHpA_400x1600 vector contained a BGHpA sequence alone and the CDSR_GFP-WPRE-BGHpA_400x1600 contained a WPRE-BGHpA sequence, the CDSR_GFP-NoBGHpA_400x1600 vector lacked both regulatory elements, thus allowing GFP expression to be controlled by the endogenous *RAG2* 3′ UTR. While the CDSR_GFP-BGHpA_400x1600 and CDSR_GFP-WPRE-BGHpA_400x1600 donors produced comparable HDR efficiencies, the CDSR_GFP-NoBGHpA_400x1600 donor induced lower HDR as observed by flow cytometry and confirmed by ddPCR (Fig. 2B, C and Supplementary Fig. 4). Interestingly, the three donors produced significantly different MFI levels, with CDSR_GFP-BGHpA_400x1600 ($2.8 \times 10^6$) being the highest followed by CDSR_GFP-WPRE-BGHpA_400x1600 and CDSR_GFP-NoBGHpA_400x1600 ($1.6 \times 10^6$ and $0.3 \times 10^6$, respectively) (Fig. 2D, E), highlighting the strength of the synthetic 3′ UTRs in modulating expression patterns (see Supplementary Note 1 and Supplementary Fig. 1 for a description of the gating strategy).

### *KI-KO* genotype engineering in HD-derived HSPCs using two-part enrichment strategy

Since *RAG2* gene regulation is critical, we aimed to fine-tune our previously published *RAG2*-correction strategy[56] by using the CDSR method. We hypothesized that replacing the entire CDS would allow for transgene expression to be driven by the *RAG2* endogenous promoter and 3′ UTR, thus enabling the transgenic dco*RAG2* cDNA expression patterns to most similarly resemble that of endogenous *RAG2*. Additionally, by replacing the entire CDS (~1.5 kb), as opposed to pushing the sequence ~4 kb downstream in the case of HDR via insertion, we are able to more closely maintain the proximity of the *RAG* genes, thus potentially conserving the ability to form the chromatin hub super-enhancer necessary for proper expression. We constructed a CDSR correction donor (herein CDSR_Corr_Endo3′UTR [Fig. 3A, Supplementary Table 1, and Supplementary Data 1]) with a 400×800 bp homology arm pattern for *KI* of the dco*RAG2* cDNA. To track the expression of dco*RAG2* cDNA and enrich for cells with successful integration, the dco*RAG2* stop codon was eliminated and replaced with a T2A self-cleaving peptide sequence followed by a truncated nerve growth factor receptor (tNGFR) reporter gene, producing in-frame transcription of the two sequences (dco*RAG2* cDNA and tNGFR). Following the translation of the fusion protein, the T2A self-cleaves producing two proteins (RAG2 and tNGFR) at a 1:1 ratio in the cell. The use of tNGFR is particularly advantageous since it enables tracking and enrichment of corrected cells and has been approved for clinical applications[65]. Additionally, we constructed two donors with synthetic 3′ UTRs following the tNGFR, one with WPRE-BGHpA

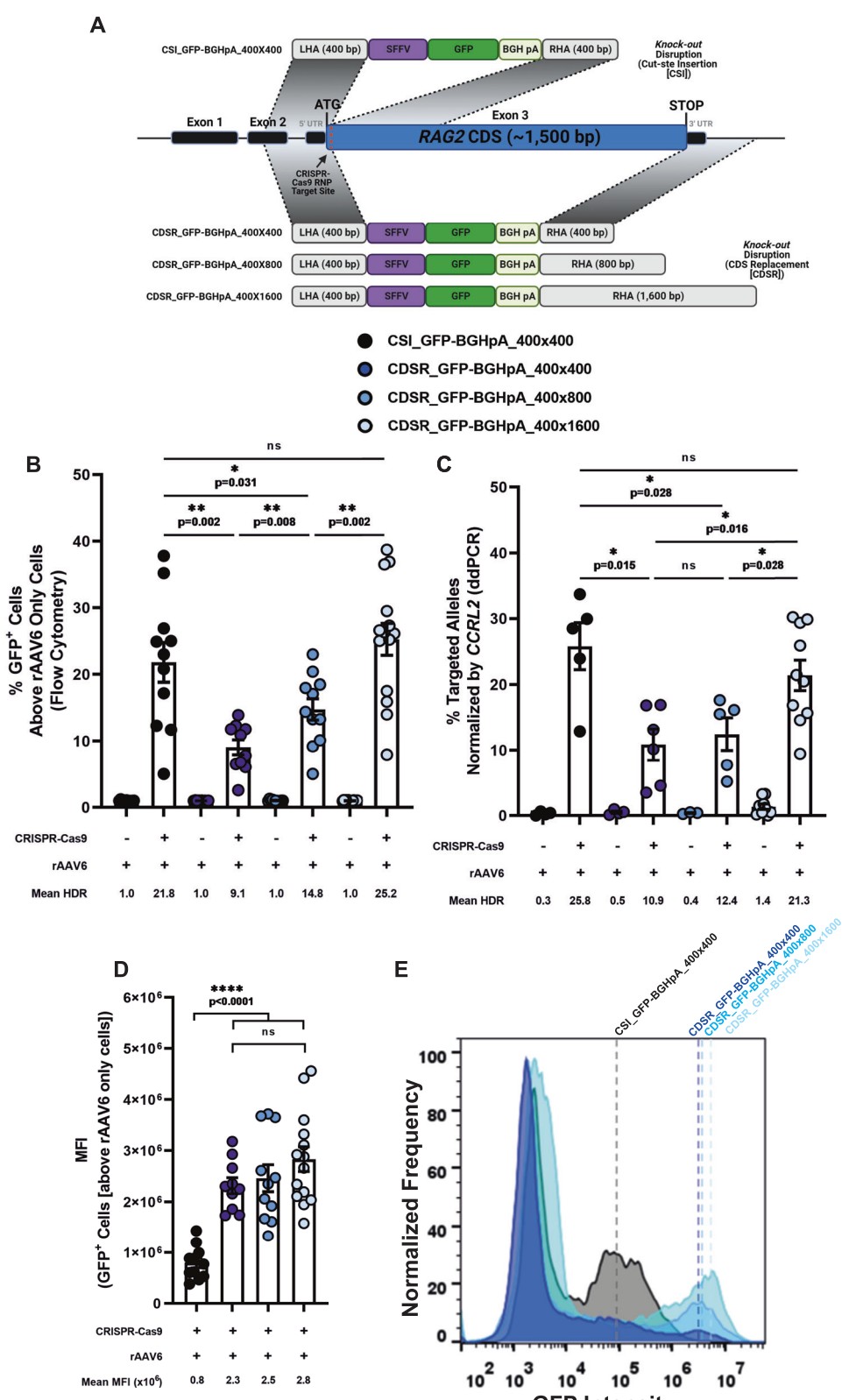

sequences and one with only the BGHpA sequence each with a 400x800 bp homology arm pattern (herein CDSR_Corr_WPRE-BGHpA and CDSR_Corr_BGHpA, respectively [Fig. 3A, Supplementary Table 1, and Supplementary Data 1]). While we observed the highest HDR for CDSR donors with a RHA of 1,600 bp (Fig. 1B, C), we could not design correction donors with a 400x1,600 bp homology arm pattern due to the limited carrying capacity (~4.8 kb) of rAAV6 vectors[66]. For

comparative purposes, we utilized our previously published CSI *KI* donor which contained dco*RAG2* cDNA followed by the *RAG2* endogenous 3' UTR sequence along with a tNGFR reporter gene cassette under the regulation of a constitutive phosphoglycerokinase (PGK) promoter and BGHpA sequence between 400 bp homology arms (herein CSI_Corr)[56] (Supplementary Fig. 5A, Supplementary Table 1, and Supplementary Data 1). We tested these four correction donors

**Fig. 1 | Different strategies for HDR: cut-site insertion vs. CDS replacement and adjusting homology arms. A** Schematic representation of *RAG2 KO* disruption donors containing a GFP reporter gene cassette under the control of an SFFV promoter and BGHpA sequence. Successful HDR of the CSI_GFP-BGHpA_400x400 donors results in the integration of the reporter gene approximately 43 bp downstream from the *RAG2* ATG start codon. Successful HDR of the three CDSR donors results in the replacement of the entire endogenous *RAG2* CDS with the reporter gene cassette. (See Table S1 for a more in-depth description of the donors and Supplementary Data 1 for the exact sequences). Figure was created with BioRender.com. **B** HDR frequencies analyzed by flow cytometry. See Supplementary Note 1 for a description of the gating strategy. CSI_GFP-BGHpA_400x400 (*N* = 11), CDSR_GFP-BGHpA_400x400 (*N* = 10), CDSR_GFP-BGHpA_400x800 (*N* = 11), CDSR_GFP-BGHpA_400×1600 (*N* = 14). Data are represented as mean ± SEM. * $p < 0.05$, ** $p < 0.01$, *** $p < 0.001$, and **** $p < 0.0001$ (Mann-Whitney one-sided test). **C** Site-specific HDR efficiencies at the *RAG2* locus measured by ddPCR and normalized by targeted *CCRL2* alleles. CSI_GFP-BGHpA_400×400 ([rAAV6 only: *N* = 4; CRISPR + AAV: *N* = 5]), CDSR_GFP-BGHpA_400×400 ([rAAV6 only: *N* = 4; CRISPR + AAV: *N* = 6]), CDSR_GFP-BGHpA_400×800 ([rAAV6 only: *N* = 3; CRISPR + AAV: *N* = 5]), CDSR_GFP-BGHpA_400×1600 ([rAAV6 only: *N* = 9; CRISPR + AAV: *N* = 10]). Data are represented as mean ± SEM. * $p < 0.05$, ** $p < 0.01$, *** $p < 0.001$, and **** $p < 0.0001$ (Mann-Whitney one-sided test test). **D** MFI values of HDR⁺ cells analyzed by flow cytometry. See Supplementary Note 1 for a description of the gating strategy. CSI_GFP-BGHpA_400×400 (*N* = 11), CDSR_GFP-BGHpA_400×400 (*N* = 10), CDSR_GFP-BGHpA_400×800 (*N* = 11), CDSR_GFP-BGHpA_400×1600 (*N* = 14). Data are represented as mean ± SEM. * $p < 0.05$, ** $p < 0.01$, *** $p < 0.001$, and **** $p < 0.0001$ (Mann-Whitney one-sided test). Source data are provided as a Source Data file. **E** Representative histograms of the MFI induced by the four *KO* donors.

individually and observed highly effective locus-specific HDR for them all, determined by ddPCR (Supplementary Fig. 5B, C). For confirmation that the integration of the donors occurred as expected, we conducted an 'in-out' PCR with one primer located on the tNGFR sequence and one primer downstream to the RHA (Supplementary Fig. 5D). Indeed, the observed amplified bands were consistent with the expected integration patterns and corroborated effective HDR (Supplementary Fig. 5E).

While our correction strategy relies on integrating the rAAV6 donor into the Cas9-induced break site via the homology recombination process, it is known that AAV donor vectors can integrate into random sites in the genome, presumed to be spontaneous DSBs, via the NHEJ pathway [67]. Additionally, the DSBs introduced by CRISPR-Cas9 at on- and off-target sites can also incorporate donor sequences in full or only partially, by NHEJ. In order to assess the specificity of the integration of our corrective donors we took advantage of ITR-seq, a highly effective method to detect integration of the rAAV6 donor's inverted terminal repeats (ITRs) across the genome [68] (ITR-seq adapters and primers can be found in Supplementary Table 3). Since any off-target integration of the donor would occur via the NHEJ repair pathway, the method is capable of detecting donor integration at any site in the genome [68]. We tested our CSI_Corr and CDSR_Corr_Endo3'UTR donors independently and found that while there was incorporation of the ITRs at the on-target site, there was relatively limited integration of ITRs the at other sites in the genome (a single off-target for each donor [Supplementary Table 4]).

Since the ITR-seq method is not quantitative and only detects sequences with ITR integration, we conducted amplification-free, long-range sequencing via Oxford Nanopore Technologies (ONT) using Cas9-targeted sequencing [69]. This method allows capturing the full scope of events, occurring upon CRISPR-Cas9-based genome editing combined with an rAAV6 donor, at the on-target locus across the cell population, without amplification bias (strategy described in the Methods section). In particular, we were interested in quantitatively assessing the extent of HDR-mediated correction versus NHEJ-based donor integration at the on-target site. Using Cas9-RNP digestion (sgRNA sequences are listed in Supplementary Table 5), we enriched for the on-target locus and analyzed the genome-editing products (Supplementary Fig. 6A, B). We found that across three replicates, the HDR frequencies determined by ONT Ampfree and the HDR frequencies determined by ddPCR (and flow cytometry in the case of CSI_Corr) were comparable (Supplementary Fig. 6C–I). Individual ONT HDR reads were determined and validated by alignment to a reference sequence displaying ideal HDR integration (Supplementary Fig. 6E, F and Supplementary Data 2). Additionally, NHEJ-based insertions to the cut site and partial NHEJ were kept below 5% and 9%, respectively for the CDSR_Corr_Endo3'UTR donor, and below 8% and 4%, respectively for the CSI_Corr donor, levels that are broadly comparable to prior reports [70–72] (Supplementary Fig. 6C–I and Supplementary Fig. 7). Lastly, we detected premature cessation of HDR when editing with the

CSI_Corr donor (4.2%), due to the presence of the 3' UTR sequence in the donor (Supplementary Fig. 6D and Supplementary Fig. 7). In these cases, the non-diverged 3' UTR sequence in the CSI_Corr donor acted as a 3' homology arm with the identical endogenous 3' UTR sequence and led to incomplete HDR.

We aimed to utilize the *KI-KO* strategy to engineer genotypes via multiplex HDR in HD-derived CD34⁺ HSPCs to simulate the therapeutic outcome of *RAG2*-SCID single-allelic correction following a gene-editing-based treatment. This strategy has two main advantages over more extensive editing methodologies: 1) In contrast to the use of induced pluripotent stem cells (iPSCs) [73–76], HD-derived CD34⁺ HSPCs are biologically authentic since they are the same cells used in HSCT; [77] and 2) Lengthy culturing protocols are inadequate since CD34⁺ HSPCs lose their regenerative ability as well as their engraftment potential after prolonged culturing [78]. Thus, we chose to apply our *KI-KO* strategy in HD-derived CD34⁺ HSPCs by utilizing multiplex HDR to obtain a cell population with one allele targeted with one of the four aforementioned correction donors and the other allele with a *KO* template (Fig. 1A [CDSR_GFP-BGHpA_400x800 was paired with the three CDSR correction donors and CSI_GFP-BGHpA_400x400 was paired with the CSI_Corr]). We utilized our *RAG2*-SCID disease model, reported on in Iancu et al., in order to compare to our correction simulation results [56].

For the CSI_Corr donor, enrichment of *KI-KO* CD34⁺ HSPCs was achieved by sorting for biallelic double-positive tNGFR⁺/GFP⁺ expression two days post-electroporation (herein day 0) and immediately seeding the cells into the in-vitro T-cell differentiation (IVTD) system (Supplementary Fig. 8A, B). However, with CDSR correction donors, since tNGFR expression is under the regulation of the endogenous *RAG2* promoter (and the *RAG2* expression window occurs later in the T-cell developmental process), there is no expression of tNGFR on day 0. Therefore, a unique enrichment strategy was required to isolate *KI-KO* cells. On day 0, the CDSR donors were sorted only for *KO* GFP expression, and the GFP⁺ cells were immediately seeded into the IVTD system. On day 14 of IVTD, when *RAG2* is highly expressed (thus, tNGFR is also expressed [Supplementary Fig. 8C]) tNGFR⁺ cells were sorted for and seeded back into the IVTD system (Fig. 3B, Supplementary Fig. 8D, and Supplementary Fig. 9). Additionally, all samples were sorted for CD7 expression to enrich for cells that have begun to differentiate, namely, only cells that were CD7⁺ were subjected to days 14-28 of IVTD (Supplementary Fig. 9). ddPCR was performed on genomic DNA from the *KI-KO* populations to confirm that the two-step enrichment method indeed led to the enrichment of a cell population with ~100% edited alleles. Indeed, in all four correction donor multiplex HDR combinations, ~100% of targeted alleles were found to be positive (Fig. 3C and Supplementary Fig. 8E).

Additionally, since the CDSR donors produce a fusion protein separated by a self-cleaving T2A sequence resulting in a 1:1 ratio between transgenic RAG2 and tNGFR, tNGFR MFI measurement was used as a proxy measurement for transgenic dco*RAG2* expression levels. As expected, we observed that the MFI for CDSR_Corr_WPRE-

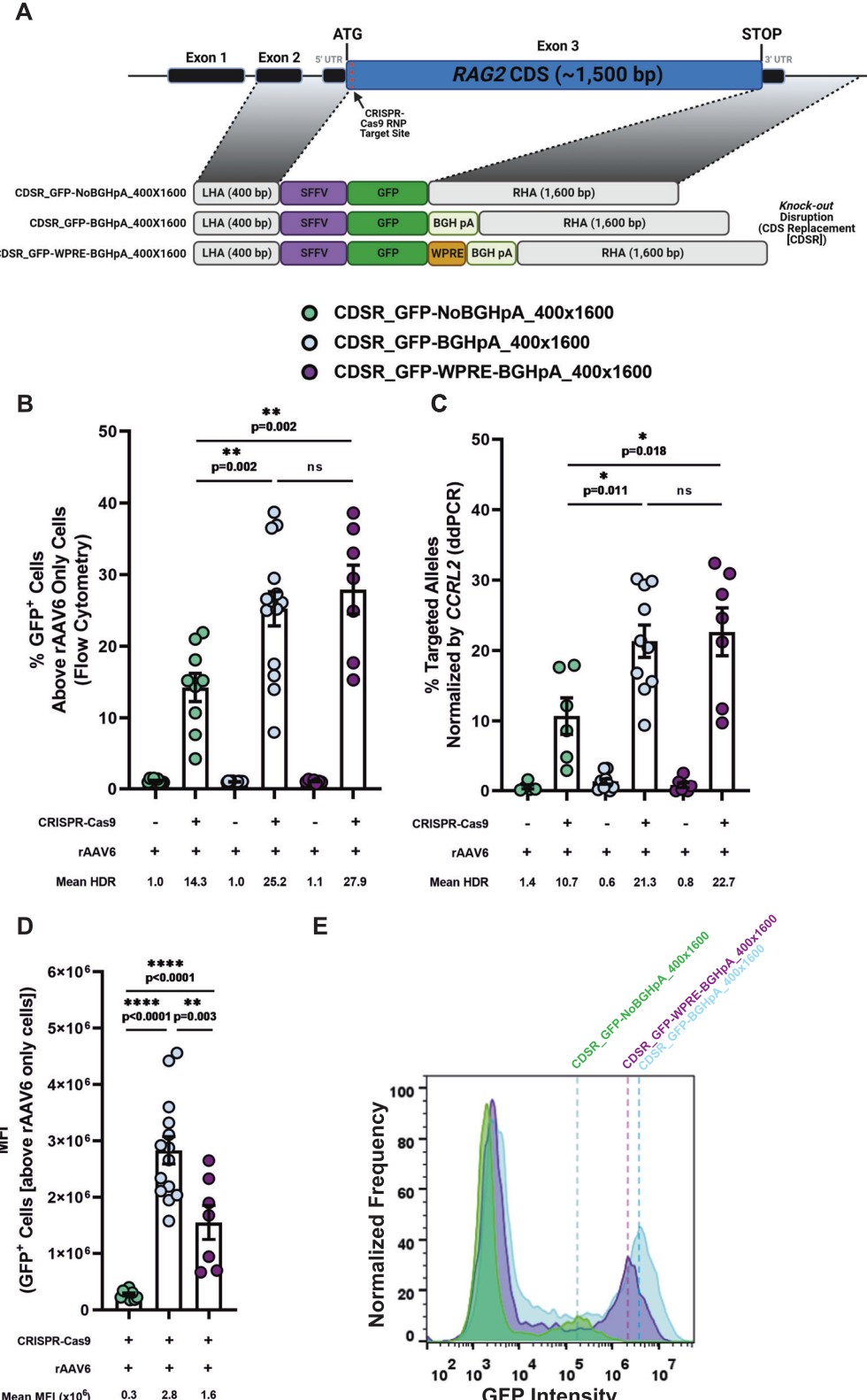

BGHpA was 2x that of CDSR_Corr_Endo3'UTR and 1.4x greater than CDSR_Corr_BGHpA on day 28 of IVTD with a similar trend on day 14 (Fig. 3D, E) indicating higher expression of the dco*RAG2*.

### KI-KO HSPCs Produce CD3$^+$TCRγδ$^+$ and CD3$^+$TCRαβ$^+$ T Cells in the IVTD System

With a robust method to isolate cells with the *KI-KO* genotype, we aimed to validate that specifically the expression of dco*RAG2* enabled

the *KI-KO* cells to differentiate into CD3$^+$ T cells with diverse TCR repertoires, to present a proof-of-concept for gene correction. Quantitative real-time PCR (qRT-PCR) using transcript-specific primer pairs (Supplementary Fig. 10A) revealed that the expression of endogenous *RAG2* was ostensibly eliminated in the *KI-KO* populations while robust dco*RAG2* cDNA expression was found exclusively in the *KI-KO* engineered cells (Fig. 4A and Supplementary Fig. 10B). Additionally, when we directly compared the total *RAG2* mRNA levels between all groups,

**Fig. 2 | Effect of synthetic polyA and/or cis-acting PREs on transgene expression. A** Schematic representation of *RAG2 KO* disruption donors containing a GFP reporter gene cassette under the control of an SFFV promoter to test the effect of different formulations of the 3′ UTR. *(Top to bottom)* CDSR_GFP-NoBGH-pA_400x1600, CDSR_GFP-BGHpA_400x1600 (data presented in Fig. 1A), and CDSR_GFP-WPRE-BGHpA_400x1600. Successful HDR of the three donors results in the replacement of the entire endogenous *RAG2* CDS with the reporter gene cassette. (See Supplementary Table 1 for a more in-depth description of the donors and Supplementary Data 1 for the exact sequences). Figure was created with BioRender.com. **B** HDR frequencies analyzed by flow cytometry. See Supplementary Note 1 for a description of the gating strategy. CDSR_GFP-NoBGHpA_400x1600 ($N = 9$), CDSR_GFP-BGHpA_400x1600 ($N = 14$), and CDSR_GFP-BGHpA_400x1600 ($N = 7$). Data are represented as mean ± SEM. * $p < 0.05$, ** $p < 0.01$, *** $p < 0.001$, and **** $p < 0.0001$ (Mann-Whitney one-sided test). **C** Site-specific HDR efficiencies at the *RAG2* locus measured by ddPCR and normalized by targeted *CCRL2* alleles. CDSR_GFP-NoBGHpA_400x1600 ([rAAV6 only: $N = 5$; CRISPR + AAV: $N = 6$]), CDSR_GFP-BGHpA_400x1600 ([rAAV6 only: $N = 9$; CRISPR + AAV: $N = 10$]), and CDSR_GFP-WPRE-BGHpA_400x1600 ([rAAV6 only: $N = 7$; CRISPR + AAV: $N = 7$]). Data are represented as mean ± SEM. * $p < 0.05$, ** $p < 0.01$, *** $p < 0.001$, and **** $p < 0.0001$ (Mann-Whitney one-sided test). **D** MFI values of HDR⁺ cells analyzed by flow cytometry. See Supplementary Note 1 for a description of the gating strategy. CDSR_GFP-NoBGHpA_400x1600 ($N = 9$), CDSR_GFP-BGHpA_400x1600 ($N = 14$), and CDSR_GFP-WPRE-BGHpA_400x1600 ($N = 7$). Data are represented as mean ± SEM. * $p < 0.05$, ** $p < 0.01$, *** $p < 0.001$, and **** $p < 0.0001$ (Mann-Whitney one-sided test). Source data are provided as a Source Data file. **E** Representative histograms of the MFI induced by the three *KO* donors.

we found that expression of the dco*RAG2* transgenes does not exceed that of the *Mock* samples indicating that the transcription is still tightly controlled and that the gene is not being overexpressed (Fig. 4B).

Importantly, the expression of the dco*RAG2* cDNA indeed facilitated T-cell development highlighted by the successful differentiation of *RAG2 KI-KO* cells into CD7⁺, CD5⁺, and CD1a⁺ pre-T cells on day 14 and CD3⁺ T cells by day 28 (Fig. 4C, D and Supplementary Fig. 10C, D [gating strategy depicted in Supplementary Fig. 11]). Additionally, robust TCRγδ expression in the CD3⁺ population was observed by flow cytometry on day 28 with the CD3⁺TCRγδ⁻ cells presumed to be CD3⁺TCRαβ⁺ T cells (Fig. 4E). Lastly, PCR amplification using primers flanking the V-J regions of the TRG locus highlighted the successful recombination of *KI-KO* cells comparable to that of the *Mock* cells on day 28 (Supplementary Fig. 10E).

### Expression of dco*RAG2* cDNA induces normal TCR repertoire development

Deep-sequencing analysis of TRB and TRG recombination on day 28 revealed diverse V(D)J rearrangement repertoires in the *RAG2 KI-KO* populations following expression of the dco*RAG2* cDNA (Fig. 5A) with no significant differences in either TRB or TRG clonotypes between the *Mock* and *RAG2 KI-KO* populations as calculated by Shannon's H and Simpson's 1-D diversity indices (Fig. 5B, C). Lastly, the complementarity determining region 3 (CDR3) lengths frequency distribution was comparable in all *RAG2 KI-KO* and *Mock* populations for both the TRB and TRG sequencing (Supplementary Fig. 12). CDR3 is the region of the TCR responsible for recognizing processed antigen peptides and its sequence and length varies from one clone to another. Thus, sequencing the CDR3 regions of a cell population is used as a measurement of TCR diversity[79]. Together, these data indicate that *KI* and expression of the dco*RAG2* cDNA promotes successful V(D)J recombination, subsequent differentiation into CD3⁺TCRαβ⁺ and CD3⁺TCRγδ⁺ T cells, and the development of highly diverse TRB and TRG repertoires.

## Discussion

When it comes to highly regulated genes such as *RAG2*, a correction strategy that replaces the entire endogenous CDS may be particularly advantageous, since it provides the ability to maintain the gene's endogenous regulatory elements and safeguard the locus architecture (see Supplementary Fig. 2A–D)[53]. Previous works to correct the *RAG* genes lacked the ability to maintain endogenous regulatory and spatiotemporal elements since the integration of the transgene was either semi-random or via insertion of thousands of bp to the Cas9-induced cut site, thus distancing the *RAG* genes one from the other and potentially altering the genomic locus and hindering optimal gene expression. While the extent to which the CDSR strategy provides an improvement regarding this central issue requires additional follow-up studies, our previously reported corrective donor (referred to in this work as CSI_Corr) has two

additional shortcomings that our CDSR correction donors aimed to improve upon[56]. Firstly, as outlined in Hubbard et al., the non-diverged 3′ UTR sequence of the CSI_Corr donor can act as a 3′ homology arm with the identical endogenous 3′ UTR sequence and lead to incomplete or early cessation of HDR[80]. Indeed, when we analyzed the gene-editing products after editing with the CSI_Corr donor by Ampfree long-read sequencing we identified events where the nondiverged 3′ UTR donor sequence acted as a 3′ homology arm leading to early cessation of HDR. While Gardner et al. and Pavel-Dinu et al. were able to avoid this possibility by introducing a BGHpA or WPRE-BGHpA sequence in place of the 3′ UTR, we believe that this is an inferior solution to relying on the endogenous regulatory sequence[73,81]. Secondly, the incorporation of a complete reporter cassette in the CSI_Corr donor can have major implications on local chromatin structure and regulation. Chiefly concerning is the presence of a constitutive PGK promoter. The insertion of such an element in a genomic locus like *RAG1/2* that requires such tight regulation is a risk that would be ideal to eliminate. Our CDSR strategy eliminates both the need to incorporate the 3′ UTR to the donor (via the CDSR strategy, the expression of the transgene relies on the endogenous genomic 3′ UTR region) as well as the need to incorporate a potentially problematic constitutive PGK promoter (via the CDSR strategy, the tNGFR cassette is tied to the dco*RAG2* by separating them with a T2A self-cleaving element). This enables us not only to track *RAG2* expression but to eliminate the need for external promoters.

In our work, we show that the CDSR method is able to induce efficient expression of the dco*RAG2* cDNA without exceeding naturally occurring *RAG2* expression levels found in *Mock* samples (25-50% expression of *RAG2*). Taken together with the efficient T-cell differentiation induced by the corrective transgenes, we describe an approach to simulating genome-editing correction for the treatment of autosomal recessive inborn errors of immunity. While our approach specifically addresses *RAG2*, the methodology can be applied to any tightly controlled gene.

Interestingly, when we designed a CDSR rAAV6 donor by distancing the RHA from the cut site (CDSR_GFP-BGHpA_400x400), HDR frequencies were significantly reduced compared to the insertion donor with equivalent homology arm lengths (CSI_GFP-BGHpA_400x400). Recently, Cromer et al. showed that elongating the length of the homology arms can lead to higher rates of HDR[53]. Thus, to account for the reduced efficiencies with the CDS replacement donor, we elongated the RHA to 800 bp which induced higher HDR frequencies, however, only once we elongated the RHA to 1,600 bp were we able to completely abrogate the low HDR efficiency observed with 400 bp. We hypothesize that by increasing the length of the distal homology arm, we enhanced the possibility for recognition and subsequent incorporation of the exogenous donor template by the cellular HDR mechanism. Additionally, we observed a significant change in expression of the reporter gene between insertion and CDS

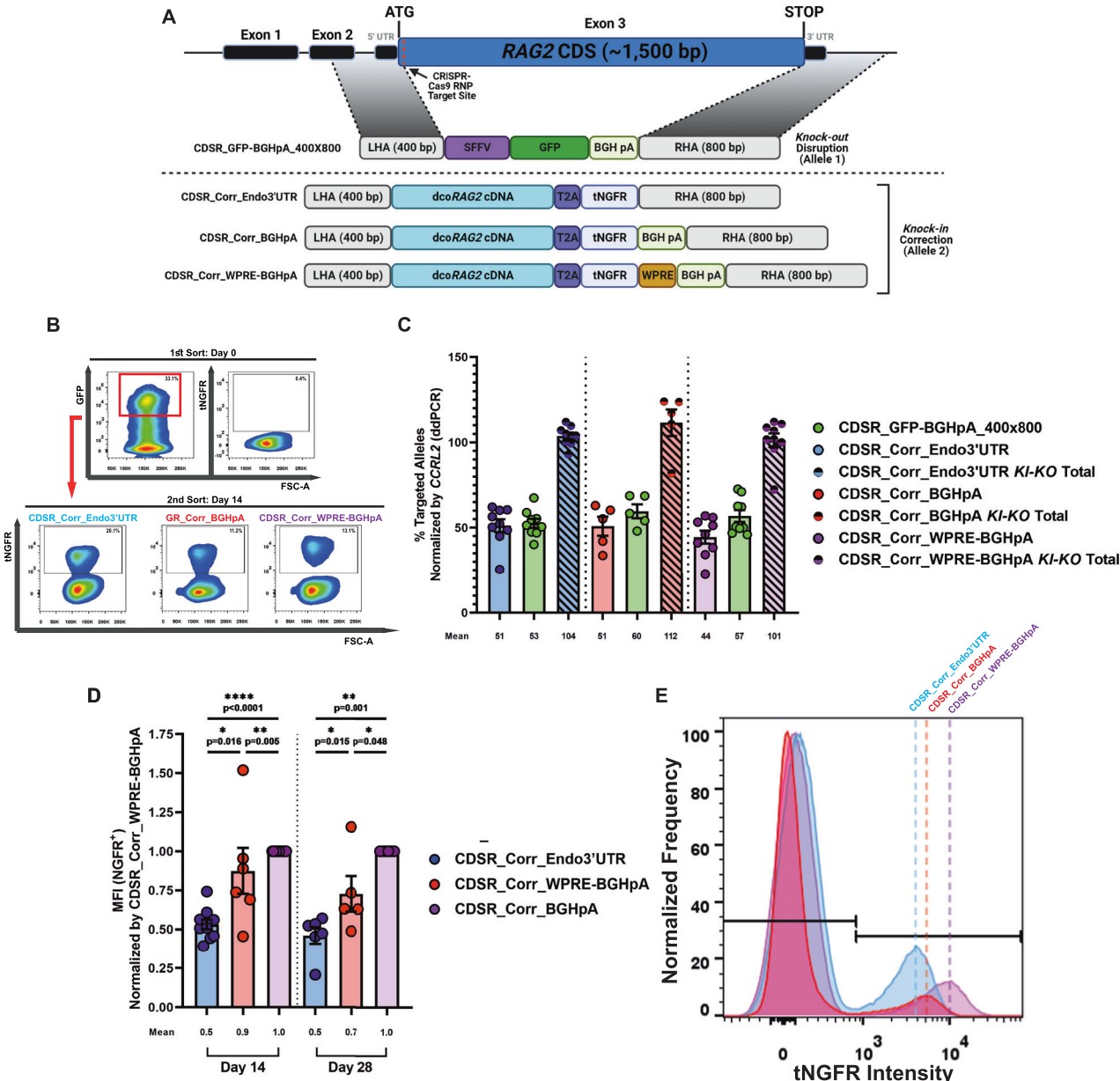

**Fig. 3 | KI-KO simulation of functional gene correction of RAG2 in HD-derived HSPCs using two-part sort strategy. A** Schematic representation of RAG2 donors for KI-KO correction simulation. (Top to bottom) CDSR_GFP-BGHpA_400×800: RAG2 disruption donor for gene KO (presented in Fig. 1A), CDSR_Corr_Endo3'UTR, CDSR_Corr_BGHpA, and CDSR_Corr_WPRE-BGHpA: RAG2 correction donors for KI of a dcoRAG2 cDNA sequence. Successful HDR of the four donors results in the replacement of the entire endogenous RAG2 CDS. (See Supplementary Table 1 for an in-depth description of the donors and Supplementary Data 1 for the exact sequences). Figure was created with BioRender.com. **B** Two-step FACS enrichment approach for KI-KO multiplexed HDR gene-targeted CD34+ HSPCs post-CRISPR-Cas9/rAAV6 editing with a combination of either CDSR_GFP-BGHpA_400×800 and CDSR_Corr_Endo3'UTR, CDSR_GFP-BGHpA_400×800 and CDSR_Corr_BGHpA, or CDSR_GFP-BGHpA_400×800 and CDSR_Corr_WPRE-BGHpA. (Top row) Representative FACS plots of the populations two days post-editing (day 0). All three correction groups express GFP yet do not express tNGFR since the RAG2 locus does not undergo transcription until later in the T-cell differentiation process.

Enrichment for GFP+ cells is conducted, and cells are seeded into the IVTD system. (Bottom row) Representative FACS plots of the populations after 14 days in IVTD (day 14). All three correction groups express tNGFR at different levels. Enrichment for tNGFR+ cells produces a homogenous double-positive tNGFR+/GFP+ population indicative of KI-KO biallelic integration. These cells are seeded into the IVTD system for another 14 days. **C** Site-specific multiplex HDR efficiencies at the RAG2 locus measured by ddPCR and normalized by targeted CCRL2 alleles. CDSR_Corr_Endo3'UTR (N = 9), CDSR_Corr_BGHpA (N = 5), and CDSR_Corr_WPRE-BGHpA (N = 10). Data are represented as mean ± SEM. **D** tNGFR MFI indicating the level of transgenic RAG2 expression on days 14 and 28 of IVTD normalized to MFI of CDSR_Corr_WPRE-BGHpA. CDSR_Corr_Endo3'UTR ([day 14: N = 10, day 28: N = 6]), CDSR_Corr_BGHpA ([day 14: N = 6, day 28: N = 5]), and CDSR_Corr_WPRE-BGHpA ([day 14: N = 11, day 28: N = 6]). Data are represented as mean ± SEM. * p < 0.05, ** p < 0.01, *** p < 0.001, and **** p < 0.0001 (Mann-Whitney one-sided test). Source data are provided as a Source Data file. **E** Representative histograms of the MFI induced by the three KI correction donors on day 14 of IVTD.

replacement donors, attributable to substantial and unique conformational changes in the edited locus which affect expression consistently over HDR at both the RAG1 and RAG2 loci. Lastly, we found that we were able to further modulate the reporter gene expression via the addition or removal of synthetic polyA sequences. Synthetic 3'

UTRs are important and easily implementable tools, some of which have been already employed in gene therapy applications[46,82]. Although these strategies proved to be effective for improving HDR and/or reporter-gene expression in the RAG2 locus, it is well documented that every genomic locus has its own unique characteristics

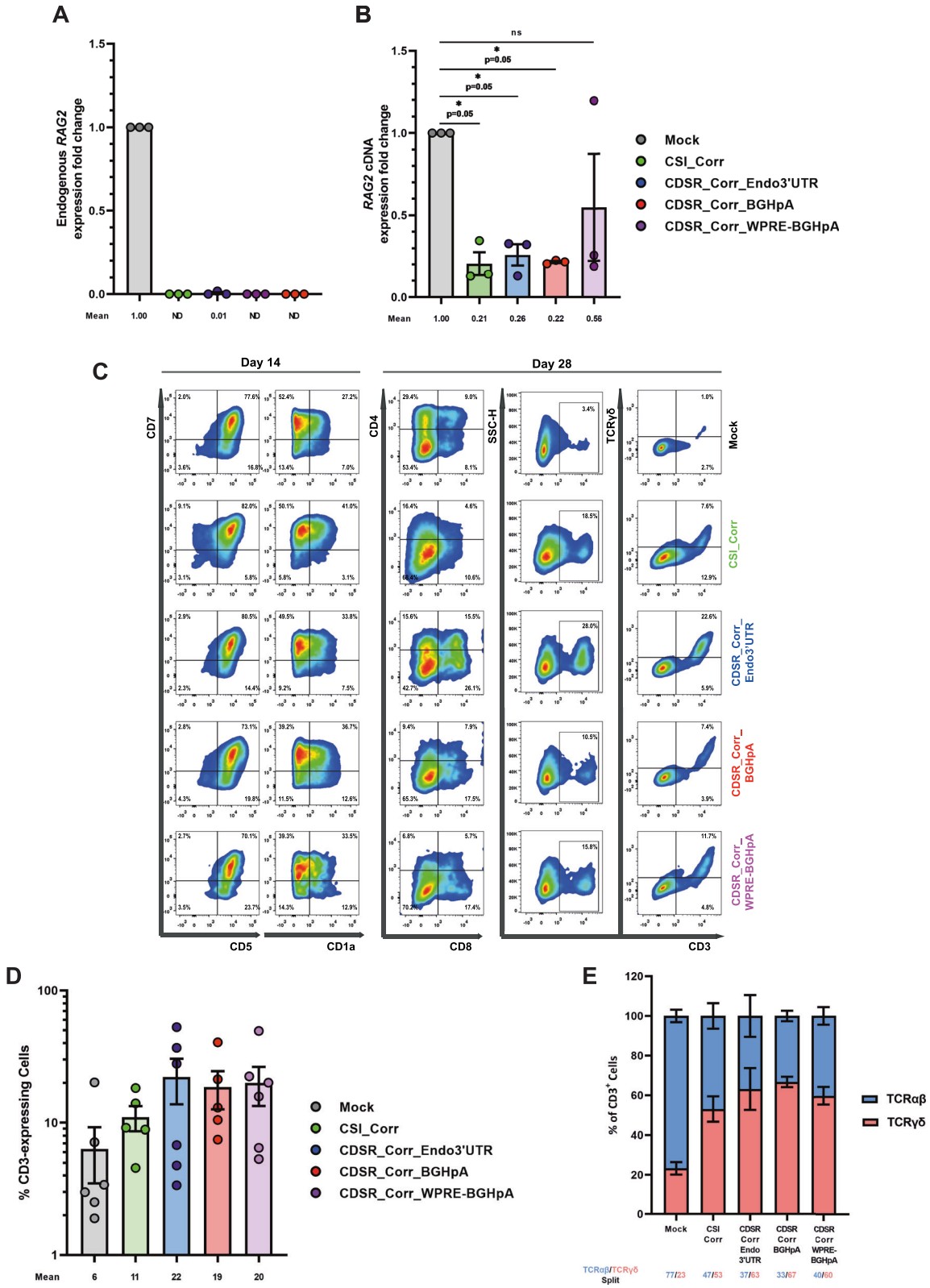

and genomic topology. Thus, we surmise that the exact length and positioning of homology arms to induce efficient HDR in both an integration and CDS replacement manner may be target- and locus-specific.

To establish that the expression of our transgene alone is capable of differentiating CD34⁺ HSPCs into CD3-expressing T cells with diverse TRB and TRG repertoires, we utilized a *KI-KO* approach for each of our CDS replacement correction donors in HD-derived CD34⁺

HSPCs. Since large amounts of SCID-patient samples are scarce, many studies have used iPSCs to develop gene-correction models[73–76]. However, in contrast to iPSCs, HD-derived CD34⁺ HSPCs provide a biologically authentic platform since they are the same cells used in HSCT[77]. The use of the *KI-KO* strategy outlined in Iancu et al. to mimic single-allelic correction of SCID-patient cells provides that all T-cell differentiative progress is due solely to the expression of the transgene and not due to the presence of endogenous *RAG2*[56]. *KI* of the transgene

**Fig. 4 | IVTD of *KI-KO* correction simulation cells produces CD3⁺TCRγδ⁺ and CD3⁺TCRαβ⁺ T cells. A** qRT-PCR quantification of endogenous *RAG2* gene expression in the *RAG2 KI-KO* cells compared to *Mock* cells on day 28 of IVTD (using green primer pair depicted in Supplementary Fig. 10A). Expression fold change is plotted relative to *Mock* and samples with no expression detected are plotted as not determined (ND). (*N* = 3 biologically independent samples). Data are represented as mean ± SEM. **B** qRT-PCR quantification of total *RAG2* expression in the *RAG2 KI-KO* cells compared to *Mock* cells on day 28 of IVTD (using red primer pair depicted in Supplementary Fig. 10A). Expression fold change is plotted relative to *Mock*. (*N* = 3 biologically independent samples). Data are represented as mean ± SEM. * *p* < 0.05, ** *p* < 0.01, *** *p* < 0.001, and **** *p* < 0.0001 (Mann-Whitney one-sided test). **C** Flow cytometry analysis of the T-cell developmental progression on days 14 and 28 of IVTD of *Mock* and *KI-KO* populations. Cells were stained for CD7, CD5, and CD1a expression on day 14 of IVTD and for CD4, CD8, CD3, and TCRγδ expression on day 28 of IVTD. Gating was determined by FMO + isotype controls. **D** Summary of CD3 expression by *Mock* and *KI-KO* populations on day 28 of IVTD. *Mock* (*N* = 6), CSI_Corr (*N* = 5), CDSR_Corr_Endo3'UTR (*N* = 6), CDSR_Corr_BGHpA (*N* = 5), and CDSR_Corr_WPRE-BGHpA (*N* = 6). Data are represented as mean ± SEM. **E** Division between TCRαβ- and TCRγδ-expressing CD3⁺ cells for *Mock* and *KI-KO* populations on day 28 of IVTD. *Mock* (*N* = 6), CSI_Corr (*N* = 5), CDSR_Corr_Endo3'UTR (*N* = 6), CDSR_Corr_BGHpA (*N* = 5), and CDSR_Corr_WPRE-BGHpA (*N* = 6). Data are represented as mean ± SEM. Source data are provided as a Source Data file.

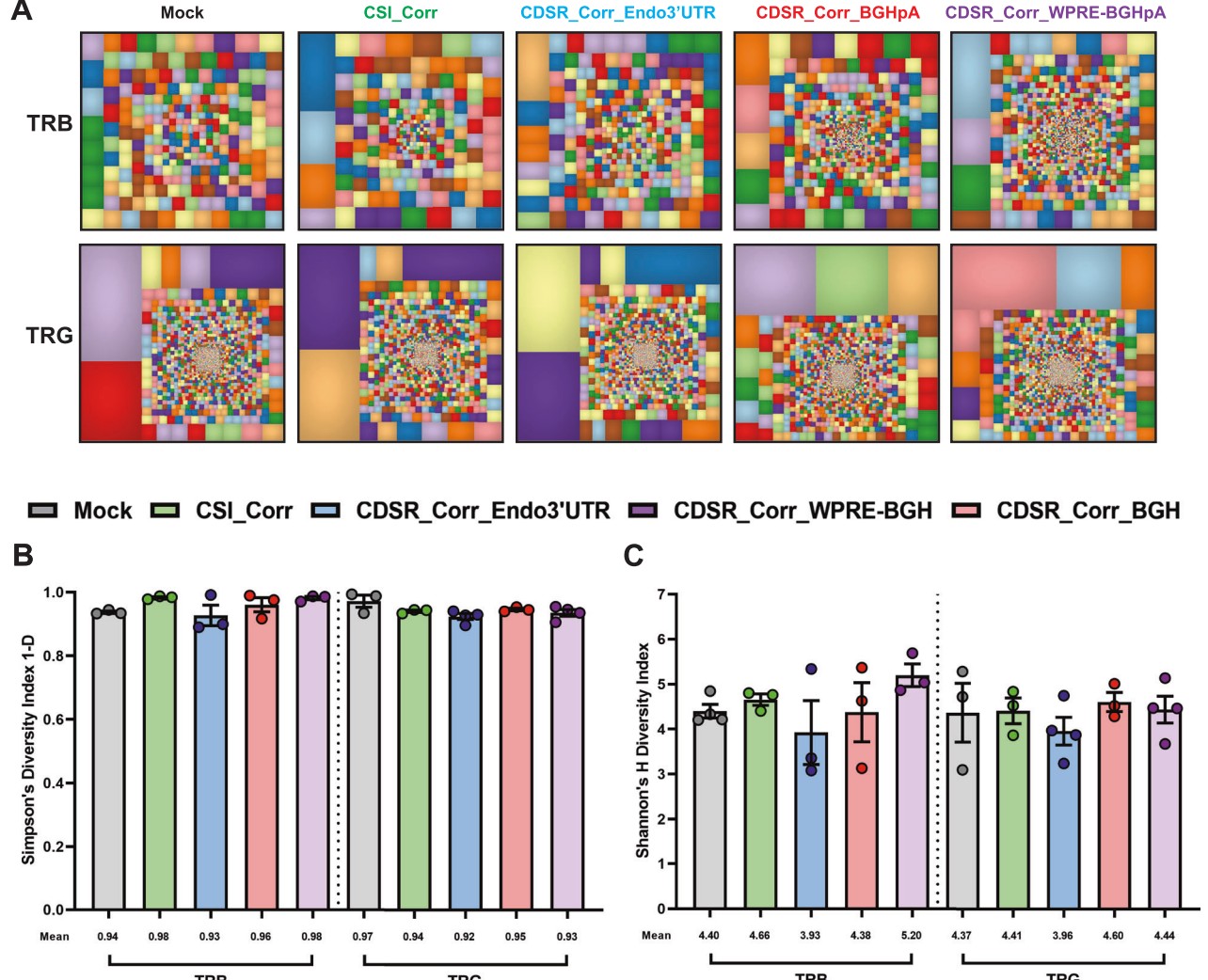

**Fig. 5 | Expression of dco*RAG2* cDNA induces normal TCRαβ and TCRγδ repertoire development. A** Representative tree map depiction of the clonal complexity of the TRB and TRG deep-sequencing repertoires of differentiated T cells from *Mock* and *KI-KO* populations. Within each treemap, each square represents a unique V-J recombination product, and the size of each square represents the clone's relative frequency. **B, C** Simpson's 1-D and Shannon's H diversity indices of TRB and TRG repertoires on day 28 of *Mock* and *KI-KO* populations. *Mock* ([TRB: *N* = 3 and TRG: *N* = 3]), CSI_Corr ([TRB: *N* = 3 and TRG: *N* = 3]), CDSR_Corr_Endo3'UTR ([TRB: *N* = 3 and TRG: *N* = 4]), CDSR_Corr_BGHpA ([TRB: *N* = 3 and TRG: *N* = 3]), and CDSR_Corr_WPRE-BGHpA ([TRB: *N* = 3 and TRG: *N* = 4]). Data are represented as mean ± SEM. Source data are provided as a Source Data file.

in one allele and reliance on Cas9-induced insertions or deletions (INDELs) for *KO* of the second allele would not be sufficient since the Cas9 does not induce *KO* INDELs in 100% of the relevant alleles (50% of the total alleles), and there would be no way to isolate the *KI-KO* cells in this case. Thus, using multiplex HDR with two distinct reporter genes to indicate the *KI-KO* genotype and FACS enrichment of the double-positive population is the most effective method for achieving ~100%

*KI-KO* enrichment in HD-derived CD34⁺ HSPCs. Since our correction donors rely on the endogenous *RAG2* promoter for expression which only begins seven days into IVTD (see Supplementary Fig. 8C), we developed a two-step enrichment process over the course of T-cell differentiation which provided a robust platform for testing our dco*RAG2* cDNA correction donors (Supplementary Fig. 9). Additionally, due to the rapid expansion of cells over the first 14 days of IVTD,

our two-step enrichment strategy enabled us to both use fewer cells and to reduce the MOI from 25,000 viral genomes (VG)/cell (used in Figs. 1 and 2) to 12,500 VG/cell while still maintaining requisite HDR. This is particularly beneficial since our group previously showed, in Allen et al., that rAAV6 vectors can trigger a potentially toxic DDR after entering cells, proportional to the MOI used[50]. Thus, reduction of the rAAV6 toxicity is required in order for a rAAV6-based method to be implemented as a clinical therapy for the purpose of gene correction. Follow-up studies to test different methods of improving HDR to allow for further reduction of the MOI, including inhibiting the NHEJ repair pathway via molecules such as i53 and/or DNA-PK inhibitors[83–85], are currently underway.

Utilizing the KI-KO strategy, we were able to track the different correction donors individually and pinpoint the effect that their expression patterns had on T-cell development. We observed higher dcoRAG2 mRNA levels in the CDS replacement correction donors than in the insertion correction samples, highlighting the effect of different gene-editing and transgene-integration strategies on the resulting gene expression. Notably, even between CDS replacement donors, we observed a significantly higher level of dcoRAG2 mRNA when the transgene expression was under the 3′ regulation of the WPRE-BGHpA sequence than the expression level under the endogenous 3′ UTR. Additionally, the RAG2 mRNA levels produced by the CDS replacement donors were observed to be 25-50% that of the Mock samples indicating that overexpression of RAG2 transgene is not occurring. We expect that the difference between mRNA levels in Mock and KI-KO samples is in part a function of only one active allele in the correction simulation samples compared to two active alleles in the Mock samples. Since parents of RAG2-SCID patients (carriers of RAG mutations ["naturally-occurring" KI-KO]) present clinically as normal and together with the fact that our KI-KO samples are able to differentiate effectively, we conclude that the mRNA levels observed are sufficient. Lastly, since the dcoRAG2 and tNGFR are expressed from the CDS replacement donors as a fusion protein, when the T2A peptide that separates the two proteins self-cleaves, the result is a 1:1 ratio of RAG2:tNGFR. Thus, the MFI of tNGFR can serve as a proxy measurement for the level of translation of RAG2 in the cell. Indeed, higher MFI was observed for the CDS replacement donor under the 3′ regulation of the WPRE-BGHpA sequence, consistent with the difference identified on the mRNA level. Together, these data indicate that the synthetic 3′ UTRs have a significant effect on the mRNA stability and/or nuclear export efficiency leading to greater translation into RAG2. Lastly, all three CDS replacement RAG2 correction donors promoted successful V(D)J recombination and subsequent differentiation into CD3+TCRαβ+ and CD3+TCRγδ+ T cells and developed highly diverse TRB and TRG repertoires. Although the CDS replacement correction donors with synthetic 3′ UTRs may be efficient for demonstrating robust transgene expression and successful T-cell development in our IVTD system, they retain the risk of leading to aberrant expression patterns in patient cells. While more extensive studies are ongoing in our lab to assess this possibility, we believe that the CDSR_Corr_Endo3′UTR holds the most promise due to its marked ability to induce CDS replacement while conserving the regulatory elements on both the 5′ and 3′ ends of the RAG2 gene. Since the 3D genomic architecture of the RAG locus is critical for proper gene expression, we expect that follow-up studies will corroborate that this strategy is the optimal one.

In summary, the aforementioned results describe a genome-editing strategy that replaces the entire RAG2 CDS (1,584 bp [in contrast to the CSI_Corr donor that leads to an introduction of an additional 3,815 bp at the Cas9 cut site]), thus treating any and all possible mutations in the gene while introducing a transgene that maintains endogenous regulatory and spatiotemporal elements. While we only present data for the two RAG genes (maximum replacement of 3,112 bp), we believe that this replacement method can be applied to other genes with a singular coding exon. Additionally, this strategy can

be employed for exon replacement where known mutations are localized on a single exon. Gray et al. showed that for a gene with multiple coding exons, the elimination of critical introns in the donor template led to a significant decrease in transgene expression[86]. Thus, selectively replacing an entire exon while retaining all critical introns could be of great importance. We believe that the CRISPR-Cas9/rAAV6-based CDS replacement strategy for a proof-of-concept gene therapy, which is site-specific and in a clinical setting would use a patient's own cells, can alleviate a number of current barriers to treatment for not only RAG2-SCID but other monogenic diseases of the blood and immune system as well. First, autologous gene therapy would eliminate the need for finding an HLA-matched donor and would reduce the risk of graft rejection. Second, using a CRISPR-Cas9/rAAV-based approach avoids the risk of dysregulated hematopoiesis, incomplete phenotypic correction, and insertional mutagenesis associated with the semi-random integration of a γRV- or LV-based approach. Third, since the 3D genomic architecture of the RAG1/2 locus is critical for proper gene expression[12,14,17,18] we were able to avoid inserting kilobases of new DNA between the two genes by using a CDS replacement strategy. This allows for transgene integration while maintaining important regulatory and spatiotemporal elements which can be crucial, especially in genes like RAG2 that are expressed in a very limited and tightly controlled window. While follow-up studies are undergoing to elucidate the potential benefits of this third point even further, here we show that expression patterns of the corrective transgene are both sufficient to induce quality T-cell differentiation while not exceeding endogenously occurring levels in the Mock samples. For these reasons, we believe that the adoption of such a strategy can assist in reducing the risks associated with gene therapy and lead to safer and more accurate applications.

## Methods

### Ethical statement
Our research complies with all relevant ethical regulations. Human cord blood (CB)-derived CD34+ HSPCs, irrespective of sex, were obtained from Sheba Medical Center CB bank under Institutional Review Board-approved protocols (Approval 3500-16-SMC). Donations of CB are collected from the obstetric delivery department after informed consent is received allowing for CB units that are not suitable for banking to be used for research purposes. No compensation is given to the donors upon consent.

### Cells and cell-culture conditions
CD34+ HSPCs were isolated via magnetic bead separation (Miltenyi Biotec). CD34+ HSPCs were cultured in SFEM II enriched with 100 ng/ml Flt3-Ligand, 100 ng/ml TPO, 100 ng/ml SCF, 0.035 mg/μl UM171, 0.75 mg/μl SR1 (Stemcell Technologies), 20 unit/ml penicillin, and 20 mg/ml streptomycin (Biological Industries, Beit Haemek, Israel) at 37 °C, 5% $CO_2$, and 5% $O_2$. Each repeat in this paper was performed on biologically unique and independent CD34+ HSPCs from different CB donors.

### CRISPR-Cas9 preparation and nucleofection
CRISPR-Cas9 RNP complex preparation and nucleofection were conducted in accordance with the extensive protocol published in Shapiro et al. [87]. A single guide RNA (sgRNA, Alt-R® sgRNA, IDT) with end chemical modifications was complexed with Cas9 at a molar ratio of 1:2.5 (Cas9:sgRNA) for 10-20 min at 25 °C to form the RNP complex. The RAG2 sgRNA variable region sequence is 5′-UGAGAAGCCUGGCU-GAAUUA-3′ and the RAG1 sgRNA variable region sequence is 5′-UUGACUCAGGGUUCCACCCA-3′. RNP complexes were added to CD34+ HSPCs reconstituted in P3 Primary Cell electroporation solution, according to the manufacturer's protocol (Lonza) at a final molar concentration of 4 μM. The cell solution was electroporated in the Lonza 4D-Nucleofector using the DZ-100 program.

## rAAV6 DNA donor design and vector production

All rAAV6 vector plasmids were designed and cloned into the pAAV-MCS plasmid containing AAV2-specific ITRs. The pDGM6 plasmid, containing the AAV6 cap genes, AAV2 rep genes, and adenovirus helper genes, was a gift from David Russell (University of Washington). The final rAAV6 vectors were produced by The University of North Carolina (UNC) Vector Core in large-scale rAAV6 batches (UNC Vector Core).

## Genome targeting and quantification

After electroporation with the *RAG2* RNP complex, CD34[+] HSPCs were seeded at a density of $0.4 \times 10^6$ cells/ml for 24 hrs. Following the incubation period, the cell density was adjusted to $0.25 \times 10^6$ cells/ml for an additional 24 hrs. In HDR experiments, the cells were transduced with the rAAV6 donor at MOIs of either 12,500 or 25,000 VG/cell within 5 min of electroporation. Cells were cultured at 37 °C, 5% $CO_2$, and 5% $O_2$ in StemSpan™ SFEM II enriched medium as noted above. Flow cytometry analyses were performed on the BD LSRFortessa™ (BD Biosciences), Aria III cell sorter (BD Biosciences), or the Accuri C6 flow cytometer (BD Biosciences). Flow cytometry analysis was performed using the FlowJo Software, Version 10.9 (https://www.flowjo.com/).

## IVTD system and immunostaining

CD34[+] HSPCs were cultured in the StemSpan™ T-Cell Generation Kit (STEMCELL Technologies, Inc.). For the first 14 days, cells were cultured in StemSpan™ SFEM II medium containing Lymphoid Progenitor Expansion Supplement in plates pre-coated with the Lymphoid Differentiation Coating Material. Cells were then harvested and re-seeded for an additional 14 days in StemSpan™ SFEM II medium containing the T-Cell Progenitor Maturation Supplement on pre-coated plates. Flow cytometry analysis was conducted on days 14 and 28 of IVTD using the LSR Fortessa™ (BD Biosciences). On day 14 of IVTD, cells were stained with PE/Cy7-anti-CD7 1:20 (clone: CD7-6B7, BioLegend), BV421-anti-CD5 1:20 (clone: UCHT2, BioLegend), PE-anti-CD1a 1:5 (clone: BL6, Beckman Coulter), and APC-anti-NGFR 1:20 (clone: ME20.4, BioLegend) antibodies. On day 28 of IVTD, cells were stained with PE/Cy7-anti-CD4 1:20 (clone: RPA-T4, BioLegend), APC-r700-anti-CD8a 1:20 (clone: RPA-T8, BD Horizon™) BV421-anti-CD3 1:20 (clone: UCHT1, BioLegend), PE-anti-TCR PAN γ/δ 1:5 (clone: IMMU510, Beckman Coulter), and APC-anti-NGFR 1:20 (clone: ME20.4, BioLegend) antibodies. BD Horizon™ Fixable Viability Stain 510 was performed on all collected cells at both time points. Gating strategies were based on fluorescence minus one (FMO) plus isotype control (at equivalent concentration to its antibody pair) samples using the following isotypes: PE/Cy7 Mouse IgG2a κ, (BioLegend), BV421 Mouse IgG1 κ (BioLegend), PE Mouse IgG1 κ (BioLegend), PE/Cy7 Mouse IgG1 κ (BioLegend), APC-r700 Mouse IgG1 κ (BD Biosciences), and APC Mouse IgG1 κ (BioLegend).

## Digital droplet PCR™ (ddPCR™)

Genomic integration quantification for HDR experiments was performed by Digital Droplet PCR™ (ddPCR™, Bio-Rad). DNA was extracted from cell populations using GeneJET Genomic DNA Purification Kit (Thermo Fisher Scientific). Each ddPCR reaction contained a HEX reference assay detecting the *CCRL2* gene to quantify the chromosome 3 copy number input[88]. FAM assays for either *KO* (disruption) or *KI* (correction) donors were designed to detect the locus-specific donor integration (assay schematics depicted in Supplementary Figs. 2F, 3E, and 5C). The ddPCR reaction was carried with the following reagents: 10 μl of ddPCR Supermix for Probes No dUTP (Bio-Rad), 1 μl each of FAM and HEX PrimeTime® Standard qPCR Assay (IDT), 1 μl restriction enzyme mix [5 μl EcoRI-HF® (NEB), 2 μl nuclease-free water, 1 μl CutSmart Buffer 10X (NEB)], genomic template DNA, and supplemented to a total of 20 μl with nuclease-free water. Droplet samples were prepared according to the manufacturer's protocol (Bio-Rad) and

40 μl of the droplet output was transferred to a 96-well plate and amplified in a Bio-Rad PCR thermocycler (Bio-Rad) at the following PCR conditions: 1 cycle at 95 °C for 10 min, then 40 cycles of 95 °C for 30 sec and 55 °C for 3 min, followed by 1 cycle at 98 °C for 10 min with a ramp rate of 2.2 °C/sec. Following the PCR, the plate was read in the QX200 Droplet Reader (Bio-Rad) and the data was analyzed using the QuantaSoft, Version 1.7, Regulatory Edition analysis software (Bio-Rad). Primer and probe sequences are listed in Table 1.

## mRNA quantification

RNA was extracted using Direct-zol™ RNA Miniprep Plus (Zymo Research) from differentiated T cells obtained on day 28 of IVTD. cDNA preparation was executed from RNA, using Oligo d(T)23 VN- S1327S (NEB), dNTPs 10 mM (Sigma-Aldrich), and M-MuLV Reverse Transcriptase (NEB). qRT-PCR reactions were conducted using the TaqMan® Fast Advanced Master Mix (Thermo Fisher Scientific) in the StepOnePlus™ Real-Time PCR System (Thermo Fisher Scientific). PCR conditions were as follows: uracil-N-glycosylase gene (UNG) incubation at 50 °C for 2 min, polymerase activation at 95 °C for 20 sec, and 40 cycles at 95 °C for 1 sec and at 60 °C for 20 sec. Primer and probe sequences are listed in Table 1.

## TRB and TRG V(D)J assessment

Genomic DNA was extracted using the GeneJET Genomic DNA Purification Kit (Thermo Fisher Scientific) from differentiated T cells on day 28 of IVTD. For TRG assessment via PCR amplification, 12 possible CDR3 clones were amplified using combinations of 4 primers for the Vγ regions and 3 primers for the Jγ regions (primer sequences are

**Table 1 | ddPCR and qRT-PCR assay sequences**

| | | |
|---|---|---|
| *RAG2 KO* (DNA) | Forward | CTCTCCCAGAGCAACAAAGAC |
| | Reverse | TTTCCATGCCTTGCAAAATGG |
| | Probe | 56-FAM/CCTACAGGT/ZEN/ GGGGTCTTTCATTCC/3IABkFQ |
| *RAG2 KI* Correction (DNA) | Forward | TCTCCCAGAGCAACAAAG |
| | Reverse | GAAGTTCATGAGGCTGAAG |
| | Probe | 56-FAM/CATAGCCTT/ZEN/AATCCAGCCCG/ 3IABkFQ |
| *RAG1 KO* (DNA) | Forward | CAGTGACTTTCAGGATGACCT |
| | Reverse | CAGCTAGCTTGCCAAACC |
| | Probe | 56-FAM/CCATGCTGG/ZEN/CTGAGGTACC/ 3IABkFQ |
| *CCRL2* (DNA) | Forward | GCTGTATGAATCCAGGTCC |
| | Reverse | CCTCCTGGCTGAGAAAAAG |
| | Probe | 5HEX/TGTTTCCTC/ZEN/CAGGA-TAAGGCAGCTGT/3IABkFQ |
| Endogenous *RAG2* (RNA) | Forward | CCAAGTGCTGACAATTAATACCTG |
| | Reverse | GACATGGTTATGCTTTACATCCAG |
| | Probe | 56-FAM/CATCAGTGA/ZEN/GAAGCCTGGCT-GAATTAAGG/3IABkFQ |
| dco*RAG2* cDNA (RNA) | Forward | CAAGTGCTGACAATTAATACCT |
| | Reverse | GAAGTTCATGAGGCTGAAG |
| | Probe | 56-FAM/CATAGCCTT/ZEN/AATCCAGCCCG/ 3IABkFQ |
| Comparison *RAG2* (RNA) | Forward | CCAAGTGCTGACAATTAATACCTG |
| | Reverse | GACATAGTTTCTGATGGTACGTAGA |
| | Probe | 56-FAM/TCACGCCTC/ZEN/ TCTGAATCTTTGCCG/3IABkFQ |
| *GAPDH* (RNA) | Forward | ACATCGCTCAGACACCATG |
| | Reverse | TGTAGTTGAGGTCAATGAAGGG |
| | Probe | 56-FAM/AAGGTCGGA/ZEN/GTCAACG-GATTTGGTC/3IABkFQ |

**Table 2 | TRG PCR amplification primers for V(D)J recombination assessment**

| | |
|---|---|
| Jγ_{1/2} | TACCTGTGACAACCAGTGTTG |
| Jγ_P | ACTTACCTGTAATGATAAGCTTT |
| Jγ_{P1/2} | TTACCAGGCGAAGTTACTATG |
| Vγ_{9-2} | ACCTGGTGAAGTCATACAGTTC |
| Vγ_{11} | CTTCCACTTCCACTTTGAAA |
| Vγ_{f1} | ACTGGTACCTACACCAGGAGG |
| Vγ_{10-2} | AGCATGGGTAAGACAAGCAA |

presented in Table 2). The PCR products were run on a 2% agarose gel. For deep sequencing of the TRB and TRG repertoires, the same genomic DNA was amplified using a multiplex master mix from either the LymphoTrack® TRB assay and/or LymphoTrack® TRG assay kits (Invivoscribe, Inc.). The amplicons were purified and sequenced using the MiSeq V2 (500 cycles) kit, with 250 bp paired-end reads (Illumina). The resulting FASTQ files were analyzed by the LymphoTrack Software −MiSeq, Version 2.4.3 (Invivoscribe, Inc.) and by the IMGT®: ImMunoGeneTics Software, Version 1 (The International ImMunoGeneTics Information System®, HighV-QUEST, http://www.imgt.org). The analysis of the incidence and clonality of TRB and TRG rearrangement sequences was performed for visual representation by the Treemap Software, Version 2019.9.1 (Macrofocus GmbH). Unique CDR3 sequence and length were determined from the total productive sequences. Lastly, Shannon's H and Simpson's 1-D diversity indices were calculated using the PAST Past4: Paleontological Statistics Software, version 4 by Øyvind Hammer[89].

### ITR-seq

ITR-seq protocol was adapted from Breton et al. [4]. Amplicons were generated from purified genomic DNA. DNA was sheared to an average size of 500 bp in 130 μl volume using an ME220 focused-ultrasonicator (Covaris, Woburn, MA), purified using AMPure beads (Beckman Coulter, Indianapolis, IN) at a 0.8x ratio, and eluted in 52 μl of IDTE pH 7.5 buffer (Integrated DNA Technologies, Coralville, Iowa, USA). End repair and dA tailing were subsequently performed in a total volume of 60 μl containing 50 μl of eluted DNA, 7 μl of ERAT buffer, and 3 μl of ERAT mix from KAPA HyperPrep Kit (cat#KK8504, Roche, Basel, Switzerland) The mix was incubated at 20 °C for 30 min, 65 °C for 30 min, and then held at 4 °C. Meanwhile, the unique Y-adapters, with molecular index tags were annealed to the common adapter (IDT) (see Supplementary Table 3 for the sequences). Each P5 adapter sequence and the Common Adapter GS were diluted to 100 μM with IDTE pH 7.5 buffer and further diluted to 30 μM with IDTE pH 7.5 buffer. Duplexes were formed in IDTE pH 7.5 for each P5 adapter sequence with the common adapter at an equimolar ratio and a final concentration of 15 μM. The final volume is according to the number of reactions needed from each P5 adapter in the ligation reaction. For annealing, the mixture was heated to 95 °C for 5 min and slowly cooled to room temperature.

After completion of the ERAT reaction, the ERAT products were ligated to the annealed adapter in the following mix: 5 μl of 10 μM annealed A01-A16 Y-adapter, 5 μl of nuclease-free water, 30 μl of Ligation buffer and 10 μl of Ligase (KAPA HyperPrep Kit, cat. no. KK8504, Roche, Basel, Switzerland), and 60 μl of the previous end-repaired and dA-tailed DNA. The ligation program was 20 °C for 15 min, and then hold at 4 °C. DNA was then purified by AMPure beads (Beckman Coulter, Indianapolis, IN) at a 0.7x ratio and eluted in 11 μl of IDTE pH 7.5 buffer. End-repaired Y-adapter-ligated DNA fragments were amplified by PCR using an ITR-specific primer and an adapter-specific primer (A01-A16_P5_FWD primer) (see Supplementary Table 3 for the sequences), in the following mix (amounts per sample): 13.4 μl

of nuclease-free water, 3 μl of 10x buffer for Taq Polymerase (MgCl_2-free, Invitrogen, Carlsbad, CA), 0.6 μl of 10 mM dNTP mix (Bio-Lab LTD, Jerusalem, Israel), 1.2 μl of 50 mM MgCl_2 (Invitrogen, Carlsbad, CA), 0.3 μl of 5 U/μl Platinum Taq polymerase (Invitrogen, Carlsbad, CA), 1 μl of 10 μM GSP_ITR3. AAV2 primer, 0.5 μl of 10 μM A01-A16_P5_FWD primer with the primer number matching the adapter number (e.g., A01_P5_FWD primer to be used with A01 Y-adapter), and 10 μl of previously purified DNA. The PCR program was 1 cycle of 95 °C for 5 min 30 cycles of 95 °C for 30 s, 69 °C for 1 min, and 72 °C for 30 s; 1 cycle at 72 °C for 5 min; and 4 °C hold. PCR products were purified using 0.7x AMPure beads (Beckman Coulter, Indianapolis, IN) and eluted in 17 μl of IDTE pH 7.5 buffer.

NGS libraries were prepared by PCR in the following mix (amounts per sample): 7.9 μl of nuclease-free water (Life Technologies, Waltham, MA), 3 μl of 10x buffer for Taq Polymerase (MgCl_2-free; Invitrogen, Carlsbad, CA), 0.6 μl of 10 mM dNTP mix (Bio-Lab LTD, Jerusalem, Israel), 1.2 μl of 50 mM MgCl_2 (Invitrogen, Carlsbad, CA), 0.3 μl of 5 U/μl Platinum Taq polymerase (Invitrogen, Carlsbad, CA), 0.5 μl of 10 μM A01-A4_P5_FWD primer with the primer number matching the adapter number, 1.5 μl of 10 μM p701−8 primers, and 15 μl of previously purified DNA (including the AMPure beads used in the previous PCR purification step). The PCR program was 1 cycle of 95 °C for 5 min; 10 cycles of 95 °C for 30 s, 75 °C for 2 min (−1 °C/cycle), and 72 °C for 30 s; 15 cycles of 95 °C for 30 s, 69 °C for 1 min, and 72 °C for 30 s; 1 cycle at 72 °C for 5 min; and 4 °C hold. PCR products were purified using 0.7x AMPure beads (Beckman Coulter, Indianapolis, IN), and resuspended in 25 μl of IDTE pH 7.5 buffer. Dual-indexed sequencing libraries were sequenced on an Illumina MiSeq V2 kit generating 2 × 150 bp paired-end reads using custom Index 1 and Read 2 primers (see Supplementary Table 3 for the sequences).

Unique molecular identifier (UMI) tagging and consolidating was performed on raw FASTQ files using the GUIDE-seq UMI module with default parameters (https://github.com/aryeelab/guideseq). R1 and R2 files were then trimmed using cutadapt with minimum-length of 15 (https://github.com/marcelm/cutadapt). Read pairs were then aligned using the bowtie2 package (https://github.com/BenLangmead/bowtie2) in a local mode against the AAV6 genome. Reads with MAPQ > 30 were further aligned to the hg38 human genome, again filtered by MAPQ > 30. Finally, genomic coordinates of validated reads (containing both human genomic sequence and fragments of the ITR sequence) were found using bedtools (https://github.com/arq5x/bedtools2).

### Cas9-targeted sequencing

Cas9 enrichment was conducted using Cas9 Sequencing Kit (cat. no. SQK-CS9109) according to the manufacturer's protocol. Two sgRNAs targeting the positive strand upstream to the region of interest, and two sgRNAs targeting the negative strand downstream to the region of interest were designed according to the Oxford Nanopore's guidelines (see Supplementary Table 5 for the sgRNA sequences). sgRNAs, Alt-R S.p. Cas9 nuclease V3, and Alt-R S.p. HiFi Cas9 nuclease V3 were purchased from IDT. The activity of RNP complexes was confirmed by digesting purified PCR amplicons produced in the target cells by specific primers and containing the target sites for the respective sgRNA, as described previously[5]. The sequencing data was generated at the Technion Genomics Center (Faculty of Medicine, Technion, Haifa, Israel) on the Oxford Nanopore MinION device (MinKNOW software v.22.12.7), using R9.4.1 flow cell (Oxford Nanopore, cat. no. FLO-MIN106D). Raw Fast5 files were base called by Guppy Basecalling Software (v6.4.8). FASTQ files were aligned to the hg38 human genome using minimap2 with default parameters[6]. Reads that were mapped to the RAG2 genomic locus were extracted and manually classified by three individuals, using Basic Local Alignment Search Tool (BLAST, https://blast.ncbi.nlm.nih.gov/Blast.cgi) as well as SnapGene software (GSL Biotech LLC, San Diego, CA, USA).

## Statistics and reproducibility

All replicates in this work were conducted on biologically unique CD34+ HSPCs from different donors. There are no technical replicates presented in these data. Statistical analyses of the data were conducted using an unpaired, one-sided Mann-Whitney test for rank comparison. We found that while the results were reproducible, as expected, donor-to-donor variation was detected in the experiments. Thus, we designed our study to have $N = 3$ or more for any given analysis. No statistical method was used to predetermine the sample size. No data was excluded unless contamination occurred over the course of the 28 days in the IVTD system. The experiments were not randomized. The Investigators were not blinded to allocation during experiments and outcome assessment.

## Reporting summary

Further information on research design is available in the Nature Portfolio Reporting Summary linked to this article.

## Data availability

The TRB and TRG sequencing data as well as the ITR-seq and ONT long-read sequencing data generated in this study have been deposited in the Sequence Read Archive (SRA) database under accession number: PRJNA926613. Source data are provided with this paper.

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

## Acknowledgements

We would like to thank the members of the Somech and the Hendel Labs for reading the manuscript and providing practical advice. Additionally, we would like to thank D. Russell for providing the pDGM6 plasmid. We give special thanks to the Technion Genomics Center team, and especially Dr. Nitsan Fourier, for excellent technical assistance and support with the long-read sequencing. This study was supported in part by research funding from the European Research Council (ERC) under the European Union Horizon 2020 research and innovation program (Grant No. 755758 A.H.) as well as the Israel Science Foundation (ISF)—Israel Precision Medicine Partnership (IPMP) (Grant No. 3115/19 A.H.) and Israel Science Foundation (ISF)—Individual Research Grants (Grant No. 2031/19 A.H.).

## Author contributions

D.A., O.K., B.I., and M.R. designed and conducted the experiments; D.A., N.K., and M.R. evaluated, and analyzed the data and performed the bioinformatics analyses, with the help and guidance of O.I. and Y.N.L.; K.B., and A.N. provided cord blood samples; K.B., Y.N.L., A.N., and R.S. critically reviewed the experiments and provided important advice; A.H. supervised and conceived the research; D.A. and A.H. wrote the manuscript, with contributions and input from all authors.

## Competing interests

The authors declare no competing interests.
