## [Peer Review File · Nature Communications]

Reviewers' Comments:

Reviewer #1:

Remarks to the Author:

The manuscript by Allen et al entitled 'CRISPR-Cas9 RAG2 Correction via Coding Sequence Replacement to Preserve Endogenous Gene Regulation and Locus Structure' describes the utilization of AAV6-based RAG2 correction as an alternative to lentiviral vector mediated RAG2 gene therapy. Although the strategy is of interest to the field the submitted study is rather rudimentary in its current form.

Major points:

- 1) Without the analysis of genome wide off-target effects in the sorted CD7 population it remains difficult to consider this strategy as a valid alternative to lentiviral gene therapy.
- 2) The title is not correct as the authors do not show correction of (pathogenic) RAG2 variants. In line with this, in vitro differentiation towards T cells is a good initial assessment of the differentiation potential but proof-of-concept can only be provided by correcting CD34+ RAG2-SCID cells and subsequent transplantation of the corrected cells into a humanized mouse model. Since RAG2-SCID patients are rare and patient-derived HSC even more rare, an alternative would be to perform an extra 'correction' experiment of introducing a pathogenic RAG2 cDNA-2A_TNGFR variant as well. In this way the in vivo T cell effects of RAG2cDNA-SCID and RAG2cDNA knockin CD34+ cells can be compared.

Other points:

- 1) Line 186: please provide a logical explanation for this difference
- 2) Targeting vectors are not described in the Methods section
- 3) dcoRAG2 cDNA is not described
- 4) dcoRAG2 cDNA is assumingly codon optimized but why do the authors introduce coRAG2 if they state in the abstract that locus conservation is key. Why not introducing regular RAG2 cDNA with some silent mutations?
- 5) Figure 4E: please speculate why the mock cells generate CD3+ cells less efficiently
- 6) Discussion: please discuss which approach is recommended for future studies

Reviewer #2:

Remarks to the Author:

Summary

In this study by Allen, Knop, and Itkowitz et al, the authors describe a gene correction strategy for RAG2-SCID. Using the now widespread HSC gene editing methodology of CRISPR/Cas9 RNP electroporation with an AAV6 HDR Template, the authors build on prior mutation-independent whole gene correction strategies by replacing Rag2's entire coding sequence at its endogenous locus with a new full length Rag2 cDNA. This approach stands in contrast to prior whole gene correction strategies which insert a new copy of the gene while leaving the now non-expressed existing mutated copy of the gene within the genome.

The HSC targeted gene editing workflow and the idea of correcting mutations by insertion of an entire gene coding sequence into its endogenous locus have been previously described, but successful gene replacement while also removing the endogenous gene has not previously been explored. This RAG2 replacement strategy enabled edited HSCs to differentiate into alpha-beta and gamma-delta T cell subtypes, as well as successfully undergo TCR gene rearrangement into a diverse repertoire. Gene replacement is a novel and useful toolset for future gene correction studies, and the successful demonstration of TCR rearrangement and differentiation in edited cells gives support to the future viability of endogenous RAG2 gene replacement strategies in the clinic.

However there is no data in the paper that shows that gene replacement has any functional difference compared to the author's earlier gene insertion method. A crucial question is thus unanswered - when would gene replacement need to be used? Either the authors should show the technical generalizability of their gene replacement method by demonstrating its effectiveness at

other loci, or they should make a functional, rather than theoretical, case as to why gene replacement at RAG2 is a better option for endogenous gene corrections.

The work overall is well done, but needs either greater technical generalization of the gene replacement strategy, or demonstration of its differential functional impact to give readers a sense for when this strategy would be warranted compared to simpler gene insertion approaches. Along with addressing the minor technical concerns outlined below, the manuscript would be suitable for publication in Nature Communications, and would be met with interest by the HSC gene editing and gene correction fields.

Major Concerns

1. Functional differences between gene replacement and gene insertion. The authors make much of the importance of endogenous regulation and spatial context of the endogenous RAG2 genomic locus, but the variable alterations to that genomic locus with the insertion vs the replacement strategies do not seem to have any impact on the tested differentiation ability of the edited HSCs (Figure 4-5). The first line of the discussion says that gene replacement is advantageous, but the presented data does not seem to make a case for this. Is the advantage of replacement just theoretical? Additional experiments and discussion as to when one might advantageously use the replacement strategy vs the insertion strategy would be needed to justify the claims presented in the abstract, introduction, and discussion.

2. Technical generalizability of gene replacement. Can gene replacement be successfully applied at other loci? How much DNA can be removed in a single gene replacement event? If gene replacement at RAG2 does not show any functional differences than gene insertion, it could be that gene replacement would show functional improvements at other loci. While the authors don't need to show functional distinctions at other sites, if there is no difference at RAG2, then it would be useful for other groups testing the gene replacement strategy at their own sites of interest to know greater technical details about the generalizability of the method.

Minor Concerns

1. The authors mention the MFI difference between the Insertion construct and Replacement constructs in Fig 1, but the gating to determine positive cells is drawn to preferentially gate the higher expressed replacement constructs. Gates in S1E seem to be underestimating the editing percentage in the CSI condition (the GFP+ population is cut in half). The majority of GFP+ cells in the Insertion condition are being cut off by the existing gates. Making the positive gate on episomally expressed cells is not a good control. A GFP negative population is the right control to draw this gate. Numbers in Fig 1B should be updated accordingly.

2. In Fig S2, "Positive" gates have been adjusted based on MFI - adjusting cutoff gates from one sample to another is iffy at best. It would be much better to use a single gate for all conditions and accurately show in the graphs in Fig 2B that there is significant episomal expression with the WPRE-BGH polyA construct.

3. ddPCR assay in 1C/2C/3C are valuable and appears consistent with the flow data, but the actual assay is not well described in results or methods. Since the data is so central to the paper, a diagram of the ddPCR assay with indications of primer/probe placement would be helpful. In contrast, the description of the "In-Out" PCR strategy in S3I is clear and helped by the diagram.

4. The presented data in 1C shows minimal detection of episomal AAV by the ddPCR assay in the Cas9 negative conditions, but what about off-target integrations of the full length AAV? Testing with a Cas9 RNP targeting a different locus would be informative as to the rate of off-target or on-target but non-HDR integration events being detected by the ddPCR assay.

5. Figure legends throughout state "N=xxx" but do not indicate whether these are technical or

biologic replicates, or a combination of the two. Should be corrected throughout, and wherever possible all experiments, especially those making claims about variability in expression levels, should include data from multiple human donors. Results text also makes no mention of whether results derive from single donor or multiple donors.

6. Insertion construct with cDNA (Figure S3) has a whole PGK-tNGFR-BGH polyA cassette inserted - the addition of this whole new expression cassette could have major implications for local chromatin structure/regulation, certainly as much or more than whether the 1.5 kb of Rag2 CDS genomic sequence is removed or not. More caveats about the difference in these construct's architectures would be warranted. An ideal insertion construct would have a T2A-tNGFR followed by an endogenous 3'UTR so that it can be directly compared to the replacement constructs.

7. Annotated DNA sequences for all AAV constructs used should be provided in supplementary data.

8. In Fig 4B, is the high variability in RAG2 gene expression with the replacement construct containing WPRE-BGHpA an inherent feature of that construct, or the result of variability between donors or technical replicates?

Reviewer #3:

Remarks to the Author:

Allen....Hendel CRISPR-Cas9 RAG2 Correction Review for Nat Comm 2023 March 7

Summary: Allen et al report a novel approach to editing the RAG2 gene for the treatment of RAG2-deficient SCID by gene therapy. Most approaches to site-specific insertion of a transgene into its endogenous locus use CRISPR Cas9 to make a double strand DNA break (DSB) in the 5' region of the gene to place the inserted gene under transcriptional control of the gene promoter. Inserting the normal transgene (coding sequence insertion-CSI) displaces the endogenous gene sequences in a 3' direction, which may alter the topological arrangement of regulatory elements and alter gene expression patterns. Instead, Allen et al use a "coding sequence replacement" (CDSR) strategy to replace the endogenous coding sequences with the transgene, which would retain the overall organization of the transcriptional domain. This is accomplished by producing RAG2 gene donor cassettes with the right homology arm matching sequences downstream from the 3' end of the endogenous gene, rather than at the 3' end of the DSB as is usually done.

By increasing the length of the right homology arm from 400 bp used in the CSI donor to 1600 bp, efficiency of targeted insertion approached but was less than that achieved by CSI with a GCP donor cassette. The effects on expression activity by the CDSR approach vs CSI was compared with donors expressing a GFP reporter gene and showed a 2.9-3.5 higher mean fluorescent intensity.

Other studies characterized the optimal 3' portion of the transgene cassette and found that adding a BHG poly A and the WPRE element each increased the level of expression, although does not address whether this may lead to some degree of abnormal expression with these exogenous elements.

Critique:

Major issues:

1. The major point of this paper, that the CDSR insertion approach will lead to more physiological expression of the RAG2 gene than CSI, is not supported by the data presented. The direct comparison between CDSR inserted vs. CSI-inserted donors expressing the clinically relevant transgene RAG2 was not done, because the method to achieve KO of one allele and knock-in to the other allele uses the CSI approach for the knock-out GFP cassette. However, this precludes making the most relevant determination of the benefits of the CDSR approach for achieving more effective transgene expression, both in vitro and in the T cell differentiation assay. In Fig 4B, the direct comparison between expression of RAG2 RNA in 28-day IVTD assay by the CSI and the CSDR cassette was not different, except for the CSDR cassette that had both the BGH and WPRE/ Production of T cells (4D, did not appear to be different with any of the donors, although this data was not analyzed statistically). The schemata of interactions of the regulatory elements of the

RAG2/RAG1 locus postulated to be affected by CSDR vs. CSI (Fig S1B) are hypothetical and there may be flexibility of the DNA that makes the effects of insertion vs replacement not important.

2. Figures 1B, 2B, show % GFP+ cells which may not be equivalent to % HDR, as the Y-axes are labelled, as there may be some off target insertions of the cassette. This is supported by the lower frequencies of target alleles measured directly by in/our PCR in concordant panels 1C and 2C.

3. There was minimal analysis of the junctions between the donors and the genomic DNA to verify that the transgene cassettes inserted properly. More in/out PCR with sequence analysis at both sides of the transgene needs to be performed, besides a single PCR that showed a band of appropriate size, which only indicates some of the insertions were appropriate.

Minor issues:

1. Panel 1A does not illustrate well that the 3' HA matches the 3' end of the RAG2 gene and may be done so by drawing dashed lines from the HA's up to the drawing of the RAG2 locus above it. It is difficult to see the colors indicating which experimental arm is being portrayed in several figures where different shades of blue are used (e.g. Fig 2 Panels B-E, S3 F).

2. Additionally, some of the labels using the light blue color are difficult to see (e.g. 1E, 2E).

3. In supplemental Figure S3, Panels G, I and J are too small to be read.

We thank the reviewers for your interest in our work and appreciate the reviewers' comments and thorough examination of our research. Based on the reviewers' comments, we have adapted figures and added data to help clarify some of the experimental conclusions.

The additions/corrections include:

- New data highlighting:
 - *RAG1* HDR targeting (Supplementary Fig. 2 and Supplementary Table 2) to emphasize the generalizability of our CDS replacement strategy to other genomic loci
 - ITR-seq specific adapters and primers (Supplementary Table 3) and off-target donor integration by ITR-seq (Supplementary Table 4) to characterize the extent of non-specific integration of our CSI and CDSR strategies
 - On-target HDR assessment by Cas9 digestion/ONT long-read sequencing (Supplementary Fig. 5 and Supplementary Table 5) to elucidate the full spectrum of editing events at the on-target sites and highlight a significant benefit to our CDSR donor over our CSI donor
 - List of all the rAAV6 donor DNA sequences (Supplementary Data 1)
- Changes in the main text addressing the reviewers' suggestions
 - All figure numbers, reference numbers, and legends have been adapted to account for the new figures and references.
 - Two new authors were added to the manuscript who had an instrumental role in conducting and analyzing the experiments needed for our thorough revision

We believe that these changes have significantly improved our manuscript. We thank you for your consideration of these revisions and your assistance during this process.

Note: Reviewers' comments are in *black*, our response is in *blue*, and text taken from the manuscript is in *red*.

Sincerely,

Ayal Hendel, Ph.D.

Reviewer comments point-by-point letter:

Reviewer #1 (Remarks to the Author):

The manuscript by Allen et al entitled ‘CRISPR-Cas9 RAG2 Correction via Coding Sequence Replacement to Preserve Endogenous Gene Regulation and Locus Structure’ describes the utilization of AAV6-based RAG2 correction as an alternative to lentiviral vector mediated RAG2 gene therapy. Although the strategy is of interest to the field the submitted study is rather rudimentary in its current form.

We thank the reviewer for their helpful feedback, and we hope that we adequately addressed their concerns regarding the thoroughness of our work below.

Major points:

1) Without the analysis of genome wide off-target effects in the sorted CD7 population it remains difficult to consider this strategy as a valid alternative to lentiviral gene therapy.

We thank Reviewer #1 for this suggestion and in order to assess the specificity of the integration of our corrective donor we ran long-read ONT sequencing on the on-target site as well as ITR-seq to detect integration of our donor at off-target sites in the genome in CD34⁺ HSPCs. Since all progenitors stem from CD34⁺ HSPCs, together with the fact that assessing the integration patterns before differentiation gives us the unbiased range of donor integration, we analyzed the DNA extracted from CD34⁺ HSPCs. Breton *et al.* (PMID: 32183699) describe a highly effective method to detect genome-wide insertions of inverted terminal repeats (ITRs) mainly derived from the nuclease's on- and off-target activity. Since any off-target integration in the genome would occur

via the NHEJ repair pathway, the method is capable of detecting integration of the donor at any site in the genome.

We tested our CSI_Corr and CDSR_Corr_Endo3'UTR donors independently and found that while there was incorporation of the ITRs at the on-target site, there has been relatively limited NHEJ-based integration of the ITRs at other sites in the genome (a single off-target for each donor [see Supplementary Table 4]). Since the ITR-seq method is not quantitative and only detects sequences with ITR integration, we sought to assess the ITR integration at the on-target site via long-read ONT sequencing after we enriched the *RAG2* locus by Cas9-RNP digestion (sgRNA sequences are listed in Supplementary Table 5). We found that across three replicates, NHEJ-based insertions to the cut site and partial NHEJ was kept below 5% and 9%, respectively for the CDSR_Corr_Endo3'UTR donor and below 8% and 4%, respectively for the CSI_Corr donor, levels that are broadly comparable to prior reports (PMID: 15208627 and 30778238 and 26948440).

We added the following text to the results section (lines 258-288) elaborating on these findings:

"While our correction strategy relies on integrating the rAAV6 donor into the Cas9-induced break site via the homology recombination process, it is known that AAV donor vectors can integrate into the random sites in the genome, presumed to be spontaneous DSBs, via the NHEJ pathway. Additionally, the DSBs introduced by CRISPR-Cas9 at on- and off-target sites can also incorporate donor sequences in full or only partially, by NHEJ. In order to assess the specificity of the integration of our corrective donors we took advantage of ITR-seq, a highly effective method to detect integration of the rAAV6 donor's inverted terminal repeats (ITRs) across the genome (strategy described in Supplementary Methods and ITR-seq adapters and primers can be found in Supplementary Table 3). Since any off-target integration of the donor would occur via the NHEJ repair pathway, the method is capable of detecting donor integration at any site in the genome. We tested our CSI_Corr and CDSR_Corr_Endo3'UTR donors independently and found that while there was incorporation

of the ITRs at the on-target site, there was relatively limited integration of the at other sites in the genome (a single off-target for each donor [Supplementary Table 4]).

Since the ITR-seq method is not quantitative and only detects sequences with ITR integration, we conducted amplification-free, long-range sequencing via Oxford Nanopore Technologies (ONT) using Cas9-targeted sequencing. This method allows capturing the full scope of events, occurring upon CRISPR-Cas9-based genome editing combined with an rAAV6 donor, at the on-target locus across the cell population, without amplification bias (strategy described in Supplementary Methods). In particular, we were interested in quantitatively assessing the extent of HDR-mediated correction versus NHEJ-based donor integration at the on-target site. Using Cas9-RNP digestion (sgRNA sequences are listed in Supplementary Table 5), we enriched for the on-target locus and analyzed the genome-editing products (Supplementary Fig. 5A-B). We found that across three replicates, the HDR frequencies determined by ONT Ampfree and the HDR frequencies determined by ddPCR (and FC in the case of CSI_Corr) were comparable (Supplementary Fig. 5C-G). Additionally, NHEJ-based insertions to the cut site and partial NHEJ were kept below 5% and 9%, respectively for the CDSR_Corr_Endo3'UTR donor, and below 8% and 4%, respectively for the CSI_Corr donor, levels that are broadly comparable to prior reports (Supplementary Fig. 5C-G). Lastly, we detect premature cessation of HDR when editing with the CSI_Corr donor (4.2%), due to the presence of the 3' UTR sequence in the donor (Supplementary Fig. 5D). In these cases, the non-diverged 3' UTR sequence in the CSI_Corr donor acts as a 3' homology arm with the identical endogenous 3' UTR sequence and leads to incomplete HDR."

Additionally, addressing the reviewer's comment regarding our strategy as a valid alternative to LV-based gene therapy, we note that CRISPR-Cas9/rAAV-based approach avoids the risk of dysregulated hematopoiesis, incomplete phenotypic correction, and insertional mutagenesis associated with the semi-random integration of a γ RV- or LV-based approach (PMID: 30664781). Thus, taken together, we concluded that our corrective donors are highly specific and when used in concert with site-specific CRISPR-Cas9 editing represent a valid alternative to LV gene therapy.

2) The title is not correct as the authors do not show correction of (pathogenic) RAG2 variants. In

line with this, in vitro differentiation towards T cells is a good initial assessment of the differentiation potential but proof-of-concept can only be provided by correcting CD34+ RAG2-SCID cells and subsequent transplantation of the corrected cells into a humanized mouse model. Since RAG2-SCID patients are rare and patient-derived HSC even more rare, an alternative would be to perform an extra 'correction' experiment of introducing a pathogenic RAG2 cDNA-2AtNGFR variant as well. In this way the in vivo T cell effects of RAG2cDNA-SCID and RAG2cDNA knockin CD34+ cells can be compared.

We appreciate the reviewer's comment, and we changed the title to "CRISPR-Cas9 RAG2 Engineering via Coding Sequence Replacement to Preserve Endogenous Gene Regulation and Locus Structure for Therapeutic Applications" in an effort to specify our innovation.

Regarding the option to introduce a pathogenic variant of *RAG2*, we believe that the data that we reported in Iancu *et al.* (PMID: 36618262) accomplishes this goal. There, we presented our *RAG2* SCID disease model, where we showed that cells with biallelic knockout of both alleles of *RAG2* did not develop into CD3⁺ T cells and were incapable of undergoing successful V(D)J recombination. We added the following reference in the text (lines 300-301) to clarify this point:

"We utilized our *RAG2*-SCID disease model, reported on in Iancu *et al.*, in order to compare our correction simulation results."

Other points:

- 1) Line 186: please provide a logical explanation for this difference

We appreciate the reviewer's comment. In the discussion we say "Additionally, we observed a significant change in expression of the reporter gene between integration and CDS replacement donors, attributable to substantial and unique conformational changes in the edited locus which

affect expression." To elaborate on this, we have included the following text (lines 201-203) in the results section:

"This difference highlights that different integration strategies have unique effects and can lead to distinctive conformational changes on the genomic locus and impact subsequent transgene expression."

2) Targeting vectors are not described in the Methods section

We appreciate Reviewer #1's note. In the Methods section, we have a section titled "rAAV6 DNA donor design and vector production" where we discuss the process of rAAV6 development as follows: "All rAAV6 vector plasmids were designed and cloned into the pAAV-MCS plasmid containing AAV2-specific inverted terminal repeats (ITRs). The pDGM6 plasmid, containing the AAV6 cap genes, AAV2 rep genes, and adenovirus helper genes, was a gift from David Russell (University of Washington). The final rAAV6 vectors were produced by The University of North Carolina (UNC) Vector Core in large-scale rAAV6 batches (UNC Vector Core)."

Additionally, we added all rAAV6 sequences in Supplemental Data 1 to clarify each donor.

3) dcoRAG2 cDNA is not described

We thank the reviewer for the comment. In the results section we described the dcoRAG2 cDNA as follows: "The dcoRAG2 cDNA produces a protein identical to WT RAG2, while the introduction of wobble changes leads to reduced similarity to the genomic sequence precluding the Cas9 from re-cutting the inserted sequence or from the inserted sequence serving as a homology arm causing premature cessation of HDR." We have moved this sentence to the introduction section where the dcoRAG2 cDNA is first referenced to clarify this point.

Additionally, we added the actual dcoRAG2 cDNA sequence in Supplemental Data 1.

- 4) dcoRAG2 cDNA is assumingly codon optimized but why do the authors introduce coRAG2 if they state in the abstract that locus conservation is key. Why not introducing regular RAG2 cDNA with some silent mutations?

The reason for diverging the cDNA and using codon optimization is because we are trying to avoid two main issues: 1) From precluding the Cas9 from re-cutting the inserted sequence (if the inserted sequence was identical to the endogenous locus, then the sgRNA could re-recognize the newly synthesized sequence and the Cas9 could recut the DNA leading to a repeated cycle that would reduce the possibility of effective HDR); or 2) from the donor sequence serving as a homology arm causing premature cessation of HDR. The dcoRAG2 cDNA is changed as little as possible from the WT sequence in order to maintain as much as possible of that original sequence while still producing the identical amino acid sequence. While there are possible epigenetic markers on the CDS that could be lost through our integration method, we must diverge this sequence in order for the complete HDR to occur. In Hubbard *et al.* (PMID: 26903548) they discuss how the 3' UTR sequence in their donor can act as a 3' homology arm and lead to incomplete or early cessation of HDR. In this case, the 3' UTR cannot be diverged since its regulatory purpose is on the RNA level (it is not translated), and the changed DNA sequence could completely alter, if not abolish, the 3' UTR's function. This is a shortcoming of our CSI_Corr donor since it incorporates the RAG2 3' UTR in the donor sequence and can lead to early cessation of HDR. Indeed, we observed via Ampfree ONT long-read sequencing that in an average of 4.23% of cases, the CSI_Corr donor's 3' UTR will act as a homology arm and lead to premature cessation of HDR (Supplementary Fig. 5). Our CDSR_Corr_Endo3'UTR donor improves on this issue by eliminating the need for incorporating the 3' UTR in the donor.

5) Figure 4E: please speculate why the mock cells generate CD3⁺ cells less efficiently

We speculate that the reason may be due to the fact that in the *KI-KO* populations, we enrich for *RAG2* expression (a feature of differentiation cells) which we cannot do in the *Mock* population. In order to isolate only *KI-KO* cells, we conduct the second round of cell sorting on day 14 of IVTD based on the expression of the corrective donor (i.e., tNGFR⁺/CD7⁺ cells [see Appendix 1 for a more thorough explanation of the experimental timeline]). In the process, we ensure that not only do we have a homogenous population of *KI-KO* cells, but that all of the cells are already robustly expressing *RAG2*. Thus, since we have a population of cells that have all begun to express *RAG2* we can deduce that the cells are all in the middle to latter stages of T-cell development and the chance for them to develop into CD3⁺ cells are marginally higher. Since the *Mock* cells are only enriched for CD7 expression and nothing can be determined about *RAG2* expression, it is possible that the sorted population is at an earlier stage of T-cell development. This may lead to a "lagging" T-cell differentiation, thus lower observed frequencies of CD3⁺ cells as determined by immunotyping on day 28 of IVTD.

Since we do not see any clonal expansion or takeover of a given clone based on the TRB and TRG sequencing in the *KI-KO* populations, nor do we observe a difference in repertoire diversity between the *KI-KO* and *Mock* populations, we do not expect any elevated risk based on the "faster" T-cell differentiation, however, further long-term follow-up studies to ensure that we are not enriching or skewing the results in a dangerous manner are undergoing.

6) Discussion: please discuss which approach is recommended for future studies

We thank the reviewer for raising this important point, and we have sought to clarify. As we have noted, overexpression of *RAG2* is a chief concern of any *RAG2* correction strategy; and restoring

RAG2 function in a manner that is as controlled as possible while still inducing the necessary phenotype is critical. We believe that the CDSR_Corr_Endo3'UTR donor provides a number of benefits over the other donors, making it the optimal choice for *RAG2* correction and future studies to elucidate its true benefits further. This donor had four main advantages:

- 1) Whereas our previously described CSI_Corr donor (PMID: 36618262) inserts into the cut site, pushing the endogenous CDS downstream and introducing kilobases of new DNA between the two *RAG* genes, the CDS replacement strategy completely swaps out the endogenous CDS, thus enabling transgene integration while maintaining important regulatory and spatiotemporal elements which can be crucial to proper gene expression. Due to the nature of the CDS replacement method, we are able to maintain the genomic architecture of the *RAG1/2* locus as much as possible and have the transgene expression driven by the regulatory elements on both the 5' and 3' ends of the *RAG2* gene. Since the 3D genomic architecture of the *RAG* locus is critical for proper gene expression this is ideal.
- 2) With the CDSR_Corr_Endo3'UTR donor, we eliminate the need to incorporate the *RAG2* 3' UTR sequence to the donor because we rely on the endogenous genomic 3' UTR region. The addition of the non-diverged 3' UTR to the CSI_Corr donor can act as a 3' homology arm and lead to incomplete or early cessation of HDR, a possibility discussed in Hubbard *et al.* (PMID: 26903548) and something that we observed via Ampfree ONT long-read sequencing occurring with the CSI_Corr donor (Supplementary Fig. 5). We sought to avoid this possibility with the CDSR_Corr_Endo3'UTR construct by eliminating the 3' UTR sequence from the donor.

- 3) With the CDSR_Corr_Endo3'UTR donor, we eliminate the need to incorporate a potentially problematic constitutive PGK promoter (the tNGFR cassette in the CSI_Corr donor is under the control of a constitutive PGK promoter). We are able to do this by tying the tNGFR cassette to the dcoRAG2 by separating them with a T2A self-cleaving element. This enables us not only to track *RAG2* expression, but to eliminate the need for external promoters with the CDSR_Corr_Endo3'UTR donor. The presence of such an element in a genomic locus like *RAG1/2* that requires such tight regulation is a risk that is ideal to eliminate.
- 4) Lastly, the CDSR_Corr_Endo3'UTR donor conserves the regulatory elements on both the 5' and 3' ends of the *RAG2* gene, contrary to the CDSR_Corr_BGHpA and CDSR_Corr_WPRE-BGHpA donors which maintained the 5' promoter region and regulatory elements but relied on synthetic 3' UTRs.

For these reasons we believe that our CDSR_Corr_Endo3'UTR donor provides a superior strategy to correct *RAG2* with lessened potential risk than the other donors; and we aim to focus future studies on understanding the full spectrum of its benefits.

We have added the following text in the discussion (lines 369-387) to make these points clearer:

"In addition to this central issue, our previously reported corrective donor (referred to in this work as CSI_Corr) has two main shortcomings that our CDSR correction donors aimed to improve upon. Firstly, as outlined in Hubbard *et al.*, the non-diverged 3' UTR sequence to the CSI_Corr donor can act as a 3' homology arm to the identical endogenous 3' UTR sequence and lead to incomplete or early cessation of HDR. Indeed, when we analyzed the gene-editing products after editing with the CSI_Corr donor by Ampfree long-read sequencing we identified events where the non-diverged 3' UTR donor sequence acted as a 3' homology arm leading to early cessation of HDR. While Gardner *et al.* and Pavel-Dinu *et al.* were able to avoid this possibility by introducing a BGHpA or a WPRE-BGHpA sequence in place of the 3' UTR, we believe that this is an inferior solution to relying on the endogenous regulatory sequence. Secondly, the incorporation of a complete reporter

cassette in the CSI_Corr donor can have major implications on local chromatin structure and regulation. Chiefly concerning is the presence of a constitutive PGK promoter. The insertion of such an element in a genomic locus like *RAG1/2* that requires such tight regulation is a risk that would be ideal to eliminate. Our CDSR strategy eliminates both the need to incorporate the 3' UTR to the donor since via replacement of the endogenous CDS, as we have the transgene rely on the endogenous genomic 3' UTR region; and the need to incorporate a potentially problematic constitutive PGK promoter by tying the tNGFR cassette to the dco*RAG2* by separating them with a T2A self-cleaving element. This enables us not only to track *RAG2* expression, but to eliminate the need for external promoters"

Reviewer #2 (Remarks to the Author):

Summary

In this study by Allen, Knop, and Itkowitz et al, the authors describe a gene correction strategy for RAG2-SCID. Using the now widespread HSC gene editing methodology of CRISPR/Cas9 RNP electroporation with an AAV6 HDR Template, the authors build on prior mutation-independent whole gene correction strategies by replacing Rag2's entire coding sequence at its endogenous locus with a new full length Rag2 cDNA. This approach stands in contrast to prior whole gene correction strategies which insert a new copy of the gene while leaving the now non-expressed existing mutated copy of the gene within the genome.

The HSC targeted gene editing workflow and the idea of correcting mutations by insertion of an entire gene coding sequence into its endogenous locus have been previously described, but successful gene replacement while also removing the endogenous gene has not previously been explored. This RAG2 replacement strategy enabled edited HSCs to differentiate into alpha-beta and gamma-delta T cell subtypes, as well as successfully undergo TCR gene rearrangement into a diverse repertoire. Gene replacement is a novel and useful toolset for future gene correction studies, and the successful demonstration of TCR rearrangement and differentiation in edited cells gives support to the future viability of endogenous RAG2 gene replacement strategies in the clinic.

However there is no data in the paper that shows that gene replacement has any functional difference compared to the author's earlier gene insertion method. A crucial question is thus unanswered - when would gene replacement need to be used? Either the authors should show the

technical generalizability of their gene replacement method by demonstrating its effectiveness at other loci, or they should make a functional, rather than theoretical, case as to why gene replacement at RAG2 is a better option for endogenous gene corrections.

The work overall is well done, but needs either greater technical generalization of the gene replacement strategy, or demonstration of its differential functional impact to give readers a sense for when this strategy would be warranted compared to simpler gene insertion approaches. Along with addressing the minor technical concerns outlined below, the manuscript would be suitable for publication in Nature Communications, and would be met with interest by the HSC gene editing and gene correction fields.

We thank the reviewer for their excellent feedback, and we hope that we adequately addressed their concerns regarding the technical generalizability of our technique and the benefits of using such a technique below.

Major Concerns

1. Functional differences between gene replacement and gene insertion. The authors make much of the importance of endogenous regulation and spatial context of the endogenous RAG2 genomic locus, but the variable alterations to that genomic locus with the insertion vs the replacement strategies do not seem to have any impact on the tested differentiation ability of the edited HSCs (Figure 4-5). The first line of the discussion says that gene replacement is advantageous, but the

presented data does not seem to make a case for this. Is the advantage of replacement just theoretical? Additional experiments and discussion as to when one might advantageously use the replacement strategy vs the insertion strategy would be needed to justify the claims presented in the abstract, introduction, and discussion.

We thank the reviewer for their important point, and we have sought to clarify. As we have noted, overexpression of *RAG2* is a chief concern of any *RAG2* correction strategy; and restoring *RAG2* function in a manner that is as controlled as possible while still inducing the necessary phenotype is critical.

Firstly, in our previous work, Iancu *et al.* (PMID: 36618262), we presented our donor CSI_Corr. This donor had three shortcomings that we sought to improve upon:

- 1) The CSI_Corr donor inserts into the cut site, pushing the endogenous CDS downstream and introducing kilobases of new DNA between the two *RAG* genes. Since the 3D genomic architecture of the *RAG* locus is critical for proper gene expression this is suboptimal.
- 2) The addition of the non-diverged 3' UTR to the CSI_Corr donor can act as a 3' homology arm and lead to incomplete or early cessation of HDR, a possibility discussed in Hubbard *et al.* (PMID: 26903548) and a phenomenon that we observed with the CSI_Corr donor via Ampfree ONT long-read sequencing. We observed that in ~4% of cases the presence of the non-diverged 3' UTR in the CSI_Corr donor leads to incomplete HDR gene-editing outcomes that do not incorporate the entire donor as intended (Supplementary Fig. 5).
- 3) The tNGFR cassette in the CSI_Corr donor is under the control of a constitutive PGK promoter. The presence of such an element in a genomic locus like *RAG1/2* that requires such tight regulation is a risk that is not ideal.

The CDSR_Corr_Endo3'UTR donor eliminates these issues as follows:

- 1) The CDS replacement strategy completely swaps out the endogenous CDS, thus enabling transgene integration while maintaining important regulatory and spatiotemporal elements which can be crucial. Due to the nature of the CDS replacement method we are able to both maintain the genomic architecture of the *RAG1/2* locus as much as possible and drive the transgene expression by regulatory elements on both the 5' and 3' ends of the *RAG2* gene.
- 2) We eliminate the need to incorporate the 3' UTR to the donor because the CDSR_Corr_Endo3'UTR donor relies on the endogenous genomic 3' UTR region. While others (Gardner *et al.* and Pavel-Dinu *et al.*) replaced the 3' UTR sequence in the donor with synthetic 3' UTR sequences to avoid this issue, we believe that eliminating 3' UTR sequences from the donor altogether and relying on the endogenous regulatory elements in the endogenous 3' UTR is a superior strategy.
- 3) We eliminate the need to incorporate a potentially problematic constitutive PGK promoter to the sensitive *RAG1/2* locus by tying the tNGFR cassette to the dco*RAG2* by separating them with a T2A self-cleaving element in the CDSR_Corr_Endo3'UTR donor. This enables us not only to track *RAG2* expression, but to eliminate the need for integration of exogenous and potentially harmful constitutive promoters.

For these reasons, we believe that our CDSR_Corr_Endo3'UTR donor provides a superior strategy to correct *RAG2* than our previously reported CSI_Corr donor.

We have added the following text in the discussion (lines 369-387) to make these points clearer:

"In addition to this central issue, our previously reported corrective donor (referred to in this work as CSI_Corr) has two main shortcomings that our CDSR correction donors aimed to improve upon. Firstly, as outlined in Hubbard *et al.*, the non-diverged 3' UTR sequence to the CSI_Corr donor can act as a 3' homology arm to the identical endogenous 3' UTR sequence and lead to incomplete or early cessation of HDR. Indeed, when we analyzed the gene-editing products after editing with the CSI_Corr donor by Ampfree long-read sequencing

we identified events where the non-diverged 3' UTR donor sequence acted as a 3' homology arm leading to early cessation of HDR. While Gardner *et al.* and Pavel-Dinu *et al.* were able to avoid this possibility by introducing a BGHpA or a WPRE-BGHpA sequence in place of the 3' UTR, we believe that this is an inferior solution to relying on the endogenous regulatory sequence. Secondly, the incorporation of a complete reporter cassette in the CSI_Corr donor can have major implications on local chromatin structure and regulation. Chiefly concerning is the presence of a constitutive PGK promoter. The insertion of such an element in a genomic locus like *RAG1/2* that requires such tight regulation is a risk that would be ideal to eliminate. Our CDSR strategy eliminates both the need to incorporate the 3' UTR to the donor since via replacement of the endogenous CDS, as we have the transgene rely on the endogenous genomic 3' UTR region; and the need to incorporate a potentially problematic constitutive PGK promoter by tying the tNGFR cassette to the dcoRAG2 by separating them with a T2A self-cleaving element. This enables us not only to track *RAG2* expression, but to eliminate the need for external promoters"

Secondly, we compared three CDS replacement donors to better understand how modulation of the dcoRAG2 transgene through synthetic 3' UTR sequences would affect T-cell differentiation. We found that while all three donors developed successfully into CD3⁺TCR $\alpha\beta$ ⁺ and CD3⁺TCR $\gamma\delta$ ⁺ T cells and developed highly diverse TRB and TRG repertoires, the expression of the dcoRAG2 transgene differed between the three. Despite detecting equivalent levels of dcoRAG2 mRNA between the three donors (Fig. 4B), we saw that the level of protein was significantly different (Fig. 3D). In line with the literature that has shown that BGHpA and WPRE-BGHpA sequences lead to increased mRNA stability and/or nuclear export efficiency, it is clear that the mRNA was more efficiently translated into RAG2 in the CDSR_Corr_BGHpA and CDSR_Corr_WPRE-BGHpA donors. While additional experiments can and will be carried out to better elucidate these differences, we believe that since all other metrics were comparable between the three donors, the CDSR_Corr_Endo3'UTR represents the donor that incurs the least risk as far as producing aberrant *RAG2* expression is concerned.

We have added/adapted the following text in the discussion (lines 462-472) to make these points clearer:

"Lastly, all three CDS replacement *RAG2* correction donors promoted successful V(D)J recombination and subsequent differentiation into CD3⁺TCRαβ⁺ and CD3⁺TCRγδ⁺ T cells and developed highly diverse TRB and TRG repertoires. Although the CDS replacement correction donors with synthetic 3' UTRs may be efficient for demonstrating robust transgene expression and successful T-cell development in our IVTD system, they retain the risk of leading to aberrant expression pattern in patient cells. While more extensive studies are ongoing in our lab to assess this possibility, we believe that the CDSR_Corr_Endo3'UTR holds the most promise due to its marked ability to induce CDS replacement while conserving the regulatory elements on both the 5' and 3' ends of the *RAG2* gene. Since the 3D genomic architecture of the *RAG* locus is critical for proper gene expression, we expect that follow-up studies will corroborate that this strategy is ideal."

2. Technical generalizability of gene replacement. Can gene replacement be successfully applied at other loci? How much DNA can be removed in a single gene replacement event? If gene replacement at *RAG2* does not show any functional differences than gene insertion, it could be that gene replacement would show functional improvements at other loci. While the authors don't need to show functional distinctions at other sites, if there is no difference at *RAG2*, then it would be useful for other groups testing the gene replacement strategy at their own sites of interest to know greater technical details about the generalizability of the method.

We thank the reviewer for this point. In the case of *RAG2*, we were able to replace 1,541bp. As noted in Fig. 1, we tried a number of different homology arm patterns in order to achieve highly efficient HDR in the CDS replacement manner. We noted in the discussion, "...we surmise that the exact length and positioning of homology arms to induce efficient HDR in both an integration and CDS replacement manner is target- and locus-specific." Taking this into account, we designed a number of rAAV6 donors that introduce a GFP expression cassette into the *RAG1* locus

(Supplementary Fig. 2A). Similar to *RAG2*, we used a highly specific sgRNA that targeted just downstream from the *RAG1* ATG start codon. Since *RAG1* CDS is longer than that of *RAG2*, the replacement method here aimed to replace 3,112bp. We found that longer homology arms were required to induce highly effective HDR at the *RAG1* locus compared to the *RAG2* locus (Supplementary Fig. 2B-C). While the most effective homology arm pattern seems to be locus-specific, we observed the same effect on MFI between the two loci, namely CDS replacement donors led to significantly higher expression of the transgene than the CSI donors (Supplementary Fig. 2D).

We added the following text to the results section (lines 191-198) to explain our results:

"To validate that our CDSR strategy is broadly applicable and not specific only to the *RAG2* locus, we designed a set of rAAV6 donors to introduce a GFP expression cassette into the *RAG1* locus (Supplementary Fig. 2A, Supplementary Table 2, and Supplementary Data 1). Similar to *RAG2*, we used a highly specific sgRNA that targeted just downstream from the *RAG1* ATG start codon. Since *RAG1* CDS is longer than that of *RAG2*, the CDSR method here replaced 3,112bp as opposed to only 1,541bp at the *RAG2* locus. While we were able to achieve highly efficient HDR at the *RAG1* locus as well, we found that longer homology arms were required to do so (Supplementary Fig. 2B-E)."

Lastly, while we only present data for the two *RAG* genes (maximum replacement of 3,112bp), we believe that this replacement method can be applied to other genes with a singular coding exon or can be employed for exon replacement where known mutations are localized on a single exon.

We added the following text to the discussion section (lines 477-483) to explain our results:

"While we only present data for the two *RAG* genes (maximum replacement of 3,112bp), we believe that this replacement method can be applied to other genes with a singular coding exon. Additionally, this strategy can be employed for exon replacement where known mutations are localized on a single exon. Gray *et al.* showed that for a gene with multiple coding exons, elimination of critical introns in the donor template led to a significant

decrease in transgene expression. Thus, selectively replacing an entire exon while retaining all critical introns could be of great importance."

Minor Concerns

1. The authors mention the MFI difference between the Insertion construct and Replacement constructs in Fig 1, but the gating to determine positive cells is drawn to preferentially gate the higher expressed replacement constructs. Gates in S1E seem to be underestimating the editing percentage in the CSI condition (the GFP⁺ population is cut in half). The majority of GFP⁺ cells in the Insertion condition are being cut off by the existing gates. Making the positive gate on episomally expressed cells is not a good control. A GFP negative population is the right control to draw this gate. Numbers in Fig 1B should be updated accordingly.

We appreciate the depth of understanding that the reviewer is looking for on this point; and we hope that we can clarify. The flow cytometry gates are determined by the level of episomal expression, consistent with the way this has been done in the literature, which varies for each individual rAAV6 donor but is estimated to be ~1% (see Bak *et al.* [PMID: 29370156]). Lowering the gate would lead to incorporation of many more cells into the calculation that are only be GFP⁺ due to this episomal expression. Thus, we utilized a control sample that was treated only with the rAVV6 donor and no CRISPR-Cas9 RNP complex for each rAVV6 donor individually. The calculated GFP⁺ values for these samples can be seen in Figs. 1A-B and 2A-B and Supplementary Figs. 1 and 2.

When calculating the MFI, if we lowered the gate to include the entire upper population or drew a uniform gate for all of the rAAV6 donors, we would be including a substantial number of cells

that are really HDR⁻ and are simply expressing the rAAV6 donor in an episomal manner. Additionally, the number of cells that would now be included, based on the uniform gate, would vary significantly between donors, thus making the calculated value meaningless (see added figure below). These figures depict the % of cells that are GFP⁺ of RNP⁻ samples if the gate was placed in a uniform manner to include the entire upper population [based on an untreated sample]. Note the significant differences in frequencies of the RNP⁻ samples indicating a differing level of episomal expression for each donor). This would lower the calculated MFI across all donors, but the general trend of CDSR donors producing a higher MFI than the CSI donors would broadly be maintained. Thus, we determined that the most logical and accurate way to determine the MFI was concurrent with the strategy employed by Cromer *et al.* (PMID: 33737751). To do so, we chose to gate each rAAV6 donor by its own RNP⁻ control and to calculate the MFI of only the cells that are determined to be HDR⁺ based on that sample's given control estimated to be ~1%.

2. In Fig S2, “Positive” gates have been adjusted based on MFI - adjusting cutoff gates from one sample to another is iffy at best. It would be much better to use a single gate for all conditions and accurately show in the graphs in Fig 2B that there is significant episomal expression with the WPRE-BGH polyA construct.

We appreciate the reviewer’s comment. As noted above, the gates are determined by the level of episomal expression which varies for each individual rAAV6 donor. What we are trying to highlight, is that the cells that are HDR⁺ express higher levels of GFP with the CDSR donors than in the CSI donors. Lowering the gate would lead to the incorporation of many more cells that are HDR⁻ that would be included in the calculation due to episomal expression of the GFP cassette. Thus, for each rAAV6 donor, we utilized a control sample that was treated only with the rAAV6 donor and no CRISPR-Cas9 RNP complex (RNP⁻ control). The values for these samples can be seen in Figs. 1A-B and 2A-B and Supplementary Figs. 1 and 2.

Again, when calculating the MFI, if we lowered the gate to include the entire upper population or drew a uniform gate for all of the rAAV6 donors, we would achieve lower overall MFI values across all donors, but the general trend of CDSR donors producing a higher MFI than the CSI donors would be broadly maintained. Additionally, as mentioned above, the number of cells that would now be included, based on a uniform gate, would vary significantly between donors, thus making the calculated value meaningless (see added figure above). Thus, we determined that the most accurate way to calculate only the MFI of HDR⁺ cells was to gate each rAAV6 donor by its own RNP⁻ control.

3. ddPCR assay in 1C/2C/3C are valuable and appears consistent with the flow data, but the actual assay is not well described in results or methods. Since the data is so central to the paper, a diagram

of the ddPCR assay with indications of primer/probe placement would be helpful. In contrast, the description of the “In-Out” PCR strategy in S3I is clear and helped by the diagram.

We thank the reviewer for this point. We have added Supplementary Figs. 1F, 2E, and 4C. These are schematics that depict the placement of the ddPCR primers for detection of site-specific *RAG2* KO targeting (Supplementary Fig. 1F), for detection of site-specific *RAG1* KO targeting (Supplementary Fig. 2E), or for detection of site-specific *RAG2* correction targeting (Supplementary Fig. 4C). Lastly, we referenced all three schematics in the Methods section on ddPCR for optimal clarity.

4. The presented data in 1C shows minimal detection of episomal AAV by the ddPCR assay in the Cas9 negative conditions, but what about off-target integrations of the full length AAV? Testing with a Cas9 RNP targeting a different locus would be informative as to the rate of off-target or on-target but non-HDR integration events being detected by the ddPCR assay.

The ddPCR assay that is used in Figure 1C and Figure 2C, are locus specific, namely they have one primer inside of the delivered donor sequence and one primer outside of the homology arm sequence inside the genomic DNA adjacent to the insertion site (see Supplementary Figs. 1F). Thus, all detection by the ddPCR is, by definition, detection of integration at the on-target site.

To address the reviewer’s comment about possible off-target integration, we ran ITR-seq to detect the integration of our donor via the NHEJ repair pathway across the genome. Breton *et al.* (PMID: 32183699) describe a highly effective method to detect insertions of inverted terminal repeats (ITRs) mainly derived from the nuclease's on- and off-target activity across the genome. We tested our CSI_Corr and CDSR_Corr_Endo3’UTR donors independently and found that while there was incorporation of the ITRs via NHEJ at the on-target site, there was relatively limited integration of

the ITRs at other sites in the genome (a single off-target for each donor [see Supplementary Table 4]). Additionally, since the ITR-seq method is not quantitative and only detects sequences with ITR integration, we sought to assess the quantitative amount of ITR integration at the on-target site via long-read ONT sequencing after we enriched the *RAG2* locus by Cas9-RNP digestion (sgRNA sequences are listed in Supplementary Table 5). We found that over three distinct replicates, NHEJ-based insertions to the cut site and partial NHEJ were kept below 5% and 9%, respectively for the CDSR_Corr_Endo3'UTR donor, and below 8% and 4%, respectively for the CSI_Corr donor (Supplementary Fig. 5C-G), levels that are broadly comparable to prior reports. (PMID: 15208627 and 30778238 and 26948440). Taken together, we concluded that our corrective donors are specific.

We added the following text to the results section (lines 258-288) elaborating on these findings:

"While our correction strategy relies on integrating the rAAV6 donor into the Cas9-induced break site via the homology recombination process, it is known that AAV donor vectors can integrate into the random sites in the genome, presumed to be spontaneous DSBs, via the NHEJ pathway. Additionally, the DSBs introduced by CRISPR-Cas9 at on- and off-target sites can also incorporate donor sequences in full or only partially, by NHEJ. In order to assess the specificity of the integration of our corrective donors we took advantage of ITR-seq, a highly effective method to detect integration of the rAAV6 donor's inverted terminal repeats (ITRs) across the genome (strategy described in Supplementary Methods and ITR-seq adapters and primers can be found in Supplementary Table 3). Since any off-target integration of the donor would occur via the NHEJ repair pathway, the method is capable of detecting donor integration at any site in the genome. We tested our CSI_Corr and CDSR_Corr_Endo3'UTR donors independently and found that while there was incorporation of the ITRs at the on-target site, there was relatively limited integration of the at other sites in the genome (a single off-target for each donor [Supplementary Table 4]).

Since the ITR-seq method is not quantitative and only detects sequences with ITR integration, we conducted amplification-free, long-range sequencing via Oxford Nanopore Technologies (ONT) using Cas9-targeted sequencing. This method allows capturing the full scope of events, occurring upon CRISPR-Cas9-based

genome editing combined with an rAAV6 donor, at the on-target locus across the cell population, without amplification bias (strategy described in Supplementary Methods). In particular, we were interested in quantitatively assessing the extent of HDR-mediated correction versus NHEJ-based donor integration at the on-target site. Using Cas9-RNP digestion (sgRNA sequences are listed in Supplementary Table 5), we enriched for the on-target locus and analyzed the genome-editing products (Supplementary Fig. 5A-B). We found that across three replicates, the HDR frequencies determined by ONT Ampfree and the HDR frequencies determined by ddPCR (and FC in the case of CSI_Corr) were comparable (Supplementary Fig. 5C-G). Additionally, NHEJ-based insertions to the cut site and partial NHEJ were kept below 5% and 9%, respectively for the CDSR_Corr_Endo3'UTR donor, and below 8% and 4%, respectively for the CSI_Corr donor, levels that are broadly comparable to prior reports (Supplementary Fig. 5C-G). Lastly, we detect premature cessation of HDR when editing with the CSI_Corr donor (4.2%), due to the presence of the 3' UTR sequence in the donor (Supplementary Fig. 5D). In these cases, the non-diverged 3' UTR sequence in the CSI_Corr donor acts as a 3' homology arm with the identical endogenous 3' UTR sequence and leads to incomplete HDR."

5. Figure legends throughout state "N=xxx" but do not indicate whether these are technical or biologic replicates, or a combination of the two. Should be corrected throughout, and wherever possible all experiments, especially those making claims about variability in expression levels, should include data from multiple human donors. Results text also makes no mention of whether results derive from single donor or multiple donors.

We appreciate the point made by the reviewer. All replicates in the entire manuscript are biological replicates; there are no technical replicates in the paper. To address this valid point, we added the following note in the Methods section (lines 508-509) to clarify this:

"Each repeat in this paper was performed on biologically unique CD34⁺ HSPCs from different CB donors."

6. Insertion construct with cDNA (Figure S3) has a whole PGK-tNGFR-BGH polyA cassette inserted - the addition of this whole new expression cassette could have major implications for

local chromatin structure/regulation, certainly as much or more than whether the 1.5 kb of Rag2 CDS genomic sequence is removed or not. More caveats about the difference in these construct's architectures would be warranted. An ideal insertion construct would have a T2A-tNGFR followed by an endogenous 3'UTR so that it can be directly compared to the replacement constructs.

We agree with the reviewer's comment and concur that better elucidation of the benefits of our CDSR donor is necessary. We agree that the addition of a completely novel expression cassette could have major implications for local chromatin structure/regulation. To specify even further, the tNGFR cassette in the CSI_Corr donor is under the control of a constitutive PGK promoter, and the presence of such an element in a genomic locus like *RAG1/2* that requires such tight regulation is a risk that would be ideal to eliminate.

To clarify, in addition to the presence of the constitutive PGK promoter, we believe that the CSI_Corr donor has two other main shortcomings that we sought to improve upon with our CDSR strategy:

- 1) The CSI_Corr donor inserts into the cut site, pushing the endogenous CDS downstream and introducing kilobases of new DNA between the two *RAG* genes. Since the 3D genomic architecture of the *RAG* locus is critical for proper gene expression this is not ideal.
- 2) The addition of the non-diverged 3' UTR to the CSI_Corr donor can act as a 3' homology arm and lead to incomplete or early cessation of HDR, a possibility discussed in Hubbard *et al.* (PMID: 26903548) and a phenomenon that we observed via Ampfree ONT long-read sequencing after editing with the CSI_Corr donor (Supplementary Fig. 5).

The CDSR_Corr_Endo3'UTR donor eliminates these three issues as follows:

- 1) We eliminate the need to incorporate a potentially problematic constitutive PGK promoter by tying the tNGFR cassette to the dcoRAG2 by separating them with a T2A self-cleaving

element. This enables us not only to track *RAG2* expression, but to eliminate the need for external exogenous and potentially problematic promoters and complete expression cassettes.

- 2) The CDS replacement strategy completely swaps out the endogenous CDS, thus enabling transgene integration while maintaining important regulatory and spatiotemporal elements which are crucial. Due to the nature of the CDS replacement method, we are able to both maintain the genomic architecture of the *RAG1/2* locus as much as possible and drive the transgene expression by regulatory element on both the 5' and 3' ends of the *RAG2* gene.
- 3) We eliminate the need to incorporate the 3' UTR sequence in the donor because we rely on the endogenous genomic 3' UTR region. While others (Gardner *et al.* and Pavel-Dinu *et al.*) replaced the 3' UTR sequence in the donor with synthetic 3' UTR sequences to avoid this issue, we believe that eliminating 3' UTR sequences from the donor altogether and relying on the endogenous regulatory elements in the endogenous 3' UTR is a superior strategy.

For these reasons, we believe that our CDSR_Corr_Endo3'UTR donor provides a superior strategy to correct *RAG2* than our previously reported CSI_Corr donor. Additionally, we believe that our data support the use of the CDSR_Corr_Endo3'UTR donor as potentially the most therapeutic approach to correct the *RAG2* gene. While the reviewer's suggestion of the additional donor could be an additional line of evidence, we believe that the proposed donor would not eliminate the issue of the 3' UTR leading to premature cessation of HDR that occurs with the CSI_Corr donor (Supplementary Fig. 5).

We have added the following texts in the discussion (lines 369-387 and lines 467-472, respectively) to make these points clearer:

"In addition to this central issue, our previously reported corrective donor (referred to in this work as CSI_Corr) has two main shortcomings that our CDSR correction donors aimed to improve upon. Firstly, as outlined in Hubbard *et al.*, the non-diverged 3' UTR sequence to the CSI_Corr donor can act as a 3' homology arm to the identical endogenous 3' UTR sequence and lead to incomplete or early cessation of HDR. Indeed, when we analyzed the gene-editing products after editing with the CSI_Corr donor by Ampfree long-read sequencing we identified events where the non-diverged 3' UTR donor sequence acted as a 3' homology arm leading to early cessation of HDR. While Gardner *et al.* and Pavel-Dinu *et al.* were able to avoid this possibility by introducing a BGHpA or a WPRE-BGHpA sequence in place of the 3' UTR, we believe that this is an inferior solution to relying on the endogenous regulatory sequence. Secondly, the incorporation of a complete reporter cassette in the CSI_Corr donor can have major implications on local chromatin structure and regulation. Chiefly concerning is the presence of a constitutive PGK promoter. The insertion of such an element in a genomic locus like *RAG1/2* that requires such tight regulation is a risk that would be ideal to eliminate. Our CDSR strategy eliminates both the need to incorporate the 3' UTR to the donor since via replacement of the endogenous CDS, as we have the transgene rely on the endogenous genomic 3' UTR region; and the need to incorporate a potentially problematic constitutive PGK promoter by tying the tNGFR cassette to the dcoRAG2 by separating them with a T2A self-cleaving element. This enables us not only to track *RAG2* expression, but to eliminate the need for external promoters"

AND

"While more extensive studies are ongoing in our lab to assess this possibility, we believe that the CDSR_Corr_Endo3'UTR holds the most promise due to its marked ability to induce CDS replacement while conserving the regulatory elements on both the 5' and 3' ends of the *RAG2* gene. Since the 3D genomic architecture of the *RAG* locus is critical for proper gene expression, we expect that follow-up studies will corroborate that this strategy is ideal."

7. Annotated DNA sequences for all AAV constructs used should be provided in supplementary data.

We appreciate the reviewer's note and have included all of the rAAV6 sequences in Supplementary Data 1.

8. In Fig 4B, is the high variability in RAG2 gene expression with the replacement construct containing WPRE-BGHpA an inherent feature of that construct, or the result of variability between donors or technical replicates?

We appreciate the reviewer's comment and hope to clarify. The introduction of additional synthetic 3' UTR sequences adds a level of variability on both mRNA stability and/or nuclear export and translation that we cannot always expect. Additionally, since we are working with biological replicates of CD34⁺ HSPCs from different subjects in each repeat, there is a donor-to-donor variation that can sometimes be unpredictable. Lastly, as we noted, this plot is N=3, thus a single outlier is very powerful. Further experiments to confirm this effect may be necessary, however, at this point, this highlights the potential risk that this donor may be capable of inducing aberrant *RAG2* expression, something we would seek to avoid at all costs.

Reviewer #3 (Remarks to the Author):

Allen....Hendel CRISPR-Cas9 RAG2 Correction Review for Nat Comm 2023 March 7

Summary: Allen et al report a novel approach to editing the RAG2 gene for the treatment of RAG2-deficient SCID by gene therapy. Most approaches to site-specific insertion of a transgene into its endogenous locus use CRISPR Cas9 to make a double strand DNA break (DSB) in the 5' region of the gene to place the inserted gene under transcriptional control of the gene promoter. Inserting the normal transgene (coding sequence insertion-CSI) displaces the endogenous gene sequences in a 3' direction, which may alter the topological arrangement of regulatory elements and alter gene expression patterns. Instead, Allen et al use a "coding sequence replacement" (CDSR) strategy to replace the endogenous coding sequences with the transgene, which would retain the overall organization of the transcriptional domain. This is accomplished by producing RAG2 gene donor cassettes with the right homology arm matching sequences downstream from the 3' end of the endogenous gene, rather than at the 3' end of the DSB as is usually done. By increasing the length of the right homology arm from 400 bp used in the CSI donor to 1600 bp, efficiency of targeted insertion approached but was less than that achieved by CSI with a GCP donor cassette. The effects on expression activity by the CDSR approach vs CSI was compared with donors expressing a GFP reporter gene and showed a 2.9-3.5 higher mean fluorescent intensity.

Other studies characterized the optimal 3' portion of the transgene cassette and found that adding a BHG poly A and the WPRE element each increased the level of expression, although does not address whether this may lead to some degree of abnormal expression with these exogenous

elements.

We thank the reviewer for their thorough feedback, and we hope that we adequately addressed their remaining concerns below.

Critique:

Major issues:

1. The major point of this paper, that the CDSR insertion approach will lead to more physiological expression of the RAG2 gene than CSI, is not supported by the data presented. The direct comparison between CDSR inserted vs. CSI-inserted donors expressing the clinically relevant transgene RAG2 was not done, because the method to achieve KO of one allele and knock-in to the other allele uses the CSI approach for the knock-out GFP cassette. However, this precludes making the most relevant determination of the benefits of the CDSR approach for achieving more effective transgene expression, both in vitro and in the T cell differentiation assay. In Fig 4B, the direct comparison between expression of RAG2 RNA in 28-day IVTD assay by the CSI and the CSDR cassette was not different, except for the CSDR cassette that had both the BGH and WPRE/ Production of T cells (4D, did not appear to be different with any of the donors, although this data was not analyzed statistically). The schemata of interactions of the regulatory elements of the RAG2/RAG1 locus postulated to be affected by CSDR vs. CSI (Fig S1B) are hypothetical and there may be flexibility of the DNA that makes the effects of insertion vs replacement not important.

We appreciate the depth of understanding that Reviewer #3 is looking for on this point and we hope that we can clarify. As we have noted, overexpression of *RAG2* is a primary concern of any

RAG2 correction strategy; and restoring *RAG2* function in a controlled manner while still inducing the necessary T-cell differentiation phenotype is critical.

Firstly, in our previous work, Iancu *et al.* (PMID: 36618262), we presented our donor CSI_Corr. This donor had three shortcomings that we sought to improve upon with our CDS replacement strategy:

- 1) The CSI_Corr donor is integrated into the cut site in an insertion manner, pushes the endogenous CDS downstream, and introduces kilobases of new DNA between the two *RAG* genes. Since the 3D genomic architecture of the *RAG* locus is critical for proper gene expression this is not ideal.
- 2) The addition of the non-diverged 3' UTR to the CSI_Corr donor can act as a 3' homology arm and lead to incomplete or early cessation of HDR, a possibility discussed in Hubbard *et al.* (PMID: 26903548) and a phenomenon that we observed with the CSI_Corr donor via Ampfree ONT long-read sequencing. We observed that incomplete integration of the CSI_Corr donor occurs ~4% of the time due to the presence of the non-diverged 3' UTR sequence in the donor that acts as a 3' homology arm (Supplementary Fig. 5).
- 3) The tNGFR cassette in the CSI_Corr donor is under the control of a constitutive PGK promoter. The presence of such an exogenous and potentially problematic element in a genomic locus like *RAG1/2* that requires such tight regulation is a risk that would be ideal to eliminate.

The CDSR_Corr_Endo3'UTR donor eliminates these issues as follows:

- 1) The CDS replacement strategy completely swaps out the endogenous CDS thus enabling transgene integration while maintaining crucially important regulatory and spatiotemporal elements. With the CDS replacement method, we are able to maintain the genomic

architecture of the *RAG1/2* locus as much as possible while having transgene expression driven by both the 5' and 3' regulatory elements of the *RAG2* gene.

- 2) We eliminate the need to incorporate the 3' UTR to the donor because we rely on the endogenous genomic 3' UTR region by replacing the entire endogenous *RAG2* CDS. While others (Gardner *et al.* and Pavel-Dinu *et al.*) replaced the 3' UTR sequence in the donor with synthetic 3' UTR sequences to avoid this issue, we believe that eliminating 3' UTR sequences from the donor altogether and relying on the endogenous regulatory elements in the endogenous 3' UTR is a superior strategy.
- 3) We eliminate the need to incorporate a potentially problematic constitutive PGK promoter by tying the tNGFR cassette to the dco*RAG2* by separating them with a T2A self-cleaving element. This enables us not only to track *RAG2* expression, but to also eliminate the need for external promoters.

For these reasons we believe that our CDSR_Corr_Endo3'UTR donor provides a superior strategy to correct *RAG2* than our previously reported CSI_Corr donor.

We have added the following paragraphs in the discussion (lines 369-387 and lines 467-472, respectively) to make these points clearer:

"In addition to this central issue, our previously reported corrective donor (referred to in this work as CSI_Corr) has two main shortcomings that our CDSR correction donors aimed to improve upon. Firstly, as outlined in Hubbard *et al.*, the non-diverged 3' UTR sequence to the CSI_Corr donor can act as a 3' homology arm to the identical endogenous 3' UTR sequence and lead to incomplete or early cessation of HDR. Indeed, when we analyzed the gene-editing products after editing with the CSI_Corr donor by Ampfree long-read sequencing we identified events where the non-diverged 3' UTR donor sequence acted as a 3' homology arm leading to early cessation of HDR. While Gardner *et al.* and Pavel-Dinu *et al.* were able to avoid this possibility by introducing a BGHpA or a WPRE-BGHpA sequence in place of the 3' UTR, we believe that this is an inferior solution to relying on the endogenous regulatory sequence. Secondly, the incorporation of a complete reporter

cassette in the CSI_Corr donor can have major implications on local chromatin structure and regulation. Chiefly concerning is the presence of a constitutive PGK promoter. The insertion of such an element in a genomic locus like *RAG1/2* that requires such tight regulation is a risk that would be ideal to eliminate. Our CDSR strategy eliminates both the need to incorporate the 3' UTR to the donor since via replacement of the endogenous CDS, as we have the transgene rely on the endogenous genomic 3' UTR region; and the need to incorporate a potentially problematic constitutive PGK promoter by tying the tNGFR cassette to the dco*RAG2* by separating them with a T2A self-cleaving element. This enables us not only to track *RAG2* expression, but to eliminate the need for external promoters"

AND

"While more extensive studies are ongoing in our lab to assess this possibility, we believe that the CDSR_Corr_Endo3'UTR holds the most promise due to its marked ability to induce CDS replacement while conserving the regulatory elements on both the 5' and 3' ends of the *RAG2* gene. Since the 3D genomic architecture of the *RAG* locus is critical for proper gene expression, we expect that follow-up studies will corroborate that this strategy is ideal."

Secondly, we compared three CDS replacement donors to better understand how modulation of the dco*RAG2* transgene through synthetic 3' UTR sequences would affect T-cell differentiation. We found that while all three donors developed successfully into CD3⁺TCR $\alpha\beta$ ⁺ and CD3⁺TCR $\gamma\delta$ ⁺ T cells and developed highly diverse TRB and TRG repertoires, the expression of the dco*RAG2* transgene significantly differed between the three donors. Despite detecting statistically equivalent levels of dco*RAG2* mRNA between the three donors (Fig. 4B), we saw that the level of protein was significantly different (Fig. 3D). Since we know that in the literature the BGHpA and WPRE-BGHpA sequences have been shown to lead to increased mRNA stability and/or nuclear export efficiency, it is clear that the mRNA was more efficiently translated into *RAG2* in the CDSR_Corr_BGHpA and CDSR_Corr_WPRE-BGHpA donors. While additional experiments can and will be carried out to better elucidate these differences, we believe that since all other

metrics were comparable between the three donors, the CDSR_Corr_Endo3'UTR represents the donor that incurs the least risk as far as producing aberrant *RAG2* expression is concerned.

We have adapted the following text in the discussion (lines 462-467) to make these points clearer:

"Lastly, all three CDS replacement *RAG2* correction donors promoted successful V(D)J recombination and subsequent differentiation into CD3⁺TCRαβ⁺ and CD3⁺TCRγδ⁺ T cells and developed highly diverse TRB and TRG repertoires. Although the CDS replacement correction donors with synthetic 3' UTRs may be efficient for demonstrating robust transgene expression and successful T-cell development in our IVTD system, they retain the risk of leading to aberrant expression pattern in patient cells."

Thirdly, since we lacked access to patient samples, we aimed to prove that expression of our dco*RAG2* cDNA alone led to a T-cell phenotype comparable to that of untreated HD-derived samples. To do so without the need for patient samples, we utilized our *KI-KO* strategy to simulate single-allelic correction of *RAG2* and isolate the effect of expression of our corrective donor by eliminating all expression of endogenous *RAG2*. We agree with the reviewer that we will need to test our donor eventually on *RAG2*-SCID patient-derived cells in order to fully validate the results from our *KI-KO* experiments, however, at the current time, these experiments are beyond the scope of our capabilities.

Lastly, as the reviewer noted, the schematic of the possible differences in genomic structure in Supplementary Fig. 1B-D are theoretical based on the literature (see Yu *et al.* [PMID: 10446057]). We want to highlight visually how the insertion of kilobases of additional DNA can potentially distort the genomic locus. Since it is known and well documented that the 3D genomic architecture of the *RAG* locus is critical for proper gene expression, this possibility is something we want to aim to alleviate as much as possible.

2. Figures 1B, 2B, show % GFP+ cells which may not be equivalent to % HDR, as the Y-axes are labelled, as there may be some off target insertions of the cassette. This is supported by the lower frequencies of target alleles measured directly by in/our PCR in concordant panels 1C and 2C.

We thank the reviewer for these points.

Firstly, we have changed the Y-axes to "% GFP+" cells to clarify that point.

Secondly, slight discrepancies between the ddPCR and the FC are expected since the ddPCR is detecting allelic integration (two alleles per cell) and the FC is detecting whole cells as only positive or negative for GFP expression. The possibility of single-allelic integration or integration without expression of GFP could lead to these minor differences.

Thirdly, to address the reviewer's concern regarding potential off-target integration, we ran ITR-seq to detect integration of our donor at off-target sites across the genome as described in Breton *et al.* (PMID: 32183699). Since any off-target integration of the donor in the genome would occur via the NHEJ repair pathway, the ITR-seq method can detect the integration of the donor's inverted terminal repeats (ITRs) at any site in the genome. We tested our CSI_Corr and CDSR_Corr_Endo3'UTR donors independently and found that while there was incorporation of the ITRs at the on-target site, there was relatively limited integration of the at other sites in the genome (a single off-target for each donor [see Supplementary Table 4]). Additionally, since the ITR-seq method is not quantitative, we sought to assess the quantitative amount of ITR integration at the on-target site via long-read ONT sequencing after we enriched the *RAG2* locus by Cas9-RNP digestion (sgRNA sequences are listed in Supplementary Table 5). We found that across three replicates, NHEJ-based insertions to the cut site and partial NHEJ was kept below 5% and 9%, respectively for the CDSR_Corr_Endo3'UTR donor and below 8% and 4%, respectively for the CSI_Corr donor, levels that are broadly comparable to prior reports (PMID: 15208627 and

30778238 and 26948440). Lastly, when we compared the on-target HDR efficiencies determined by ONT sequencing to the values obtained by FC and ddPCR, we found that they were broadly similar (Supplementary Fig. 5). While the Cas9 digestion ONT sequencing is not strictly quantitative on its own due to its low read depth, when viewed in concert with the ddPCR and FC data, we conclude that the HDR integration is not only accurate but also the efficiencies are broadly reliable.

We have added the following text to the results section (lines 258-288) to elaborate on these findings:

"While our correction strategy relies on integrating the rAAV6 donor into the Cas9-induced break site via the homology recombination process, it is known that AAV donor vectors can integrate into the random sites in the genome, presumed to be spontaneous DSBs, via the NHEJ pathway. Additionally, the DSBs introduced by CRISPR-Cas9 at on- and off-target sites can also incorporate donor sequences in full or only partially, by NHEJ. In order to assess the specificity of the integration of our corrective donors we took advantage of ITR-seq, a highly effective method to detect integration of the rAAV6 donor's inverted terminal repeats (ITRs) across the genome (strategy described in Supplementary Methods and ITR-seq adapters and primers can be found in Supplementary Table 3). Since any off-target integration of the donor would occur via the NHEJ repair pathway, the method is capable of detecting donor integration at any site in the genome. We tested our CSI_Corr and CDSR_Corr_Endo3'UTR donors independently and found that while there was incorporation of the ITRs at the on-target site, there was relatively limited integration of the at other sites in the genome (a single off-target for each donor [Supplementary Table 4]).

Since the ITR-seq method is not quantitative and only detects sequences with ITR integration, we conducted amplification-free, long-range sequencing via Oxford Nanopore Technologies (ONT) using Cas9-targeted sequencing. This method allows capturing the full scope of events, occurring upon CRISPR-Cas9-based genome editing combined with an rAAV6 donor, at the on-target locus across the cell population, without amplification bias (strategy described in Supplementary Methods). In particular, we were interested in quantitatively assessing the extent of HDR-mediated correction versus NHEJ-based donor integration at the on-target site. Using Cas9-RNP digestion (sgRNA sequences are listed in Supplementary Table 5), we enriched

for the on-target locus and analyzed the genome-editing products (Supplementary Fig. 5A-B). We found that across three replicates, the HDR frequencies determined by ONT Ampfree and the HDR frequencies determined by ddPCR (and FC in the case of CSI_Corr) were comparable (Supplementary Fig. 5C-G). Additionally, NHEJ-based insertions to the cut site and partial NHEJ were kept below 5% and 9%, respectively for the CDSR_Corr_Endo3'UTR donor, and below 8% and 4%, respectively for the CSI_Corr donor, levels that are broadly comparable to prior reports (Supplementary Fig. 5C-G). Lastly, we detect premature cessation of HDR when editing with the CSI_Corr donor (4.2%), due to the presence of the 3' UTR sequence in the donor (Supplementary Fig. 5D). In these cases, the non-diverged 3' UTR sequence in the CSI_Corr donor acts as a 3' homology arm with the identical endogenous 3' UTR sequence and leads to incomplete HDR."

3. There was minimal analysis of the junctions between the donors and the genomic DNA to verify that the transgene cassettes inserted properly. More in/out PCR with sequence analysis at both sides of the transgene needs to be performed, besides a single PCR that showed a band of appropriate size, which only indicates some of the insertions were appropriate.

To address the reviewer's comment, we conducted long-read ONT sequencing for the CSI_Corr and CDSR_Corr_Endo3'UTR donors after we enriched the *RAG2* locus by Cas9-RNP digestion (sgRNA sequences are listed in Table S4). We found that over three replicates, on-target HDR efficiencies determined by ONT sequencing to the values obtained by FC and ddPCR were broadly similar and that these reads had effective and complete integration of the donor as expected. Frequencies for the distribution of gene-editing events can be found in Supplementary Fig. 5. Additionally, NHEJ-based insertions to the cut site and partial NHEJ was kept below 5% and 9%, respectively for the CDSR_Corr_Endo3'UTR donor and below 8% and 4%, respectively for the CSI_Corr donor, levels that are broadly comparable to prior reports (PMID: 15208627 and 30778238 and 26948440). Interestingly, we observed that in ~4% of cases the presence of the non-diverged 3' UTR in the CSI_Corr donor leads to incomplete HDR gene-editing outcomes that do

not incorporate the entire donor as intended, an issue that was not relevant for the CDSR_Corr_Endo3'UTR donor. In short, in the majority of instances where editing occurred, the rAAV6 donor was incorporated in its entirety in the expected orientation adding another level of evidence to the previous in-out PCR.

We added the following text to the results section (lines 271-288) elaborating on these findings:

"Since the ITR-seq method is not quantitative and only detects sequences with ITR integration, we conducted amplification-free, long-range sequencing via Oxford Nanopore Technologies (ONT) using Cas9-targeted sequencing. This method allows capturing the full scope of events, occurring upon CRISPR-Cas9-based genome editing combined with an rAAV6 donor, at the on-target locus across the cell population, without amplification bias (strategy described in Supplementary Methods). In particular, we were interested in quantitatively assessing the extent of HDR-mediated correction versus NHEJ-based donor integration at the on-target site. Using Cas9-RNP digestion (sgRNA sequences are listed in Supplementary Table 5), we enriched for the on-target locus and analyzed the genome-editing products (Supplementary Fig. 5A-B). We found that across three replicates, the HDR frequencies determined by ONT Ampfree and the HDR frequencies determined by ddPCR (and FC in the case of CSI_Corr) were comparable (Supplementary Fig. 5C-G). Additionally, NHEJ-based insertions to the cut site and partial NHEJ were kept below 5% and 9%, respectively for the CDSR_Corr_Endo3'UTR donor, and below 8% and 4%, respectively for the CSI_Corr donor, levels that are broadly comparable to prior reports (Supplementary Fig. 5C-G). Lastly, we detect premature cessation of HDR when editing with the CSI_Corr donor (4.2%), due to the presence of the 3' UTR sequence in the donor (Supplementary Fig. 5D). In these cases, the non-diverged 3' UTR sequence in the CSI_Corr donor acts as a 3' homology arm with the identical endogenous 3' UTR sequence and leads to incomplete HDR."

Minor issues:

1. Panel 1A does not illustrate well that the 3' HA matches the 3' end of the RAG2 gene and may be done so by drawing dashed lines from the HA's up to the drawing of the RAG2 locus above it.

It is difficult to see the colors indicating which experimental arm is being portrayed in several figures where different shades of blue are used (e.g. Fig 2 Panels B-E, S3 F).

We appreciate the reviewer's note. In Fig. 2 we changed the color of the CDSR_GFP-WPRE-BGHpA_400x1600 donor to make the difference more easily noticeable.

To address the visualization of where the 3' homology arm matches up to the 3' end of the *RAG2* gene, we increased the thickness and darkness of the dashed lines and added a shaded background for emphasis.

2. Additionally, some of the labels using the light blue color are difficult to see (e.g. 1E, 2E).

We appreciate the reviewer's note. In Figs. 1 and 2 we edited the colors to brighter colors to make them easier to see.

3. In supplemental Figure S3, Panels G, I and J are too small to be read.

We appreciate the reviewer's note and have rearranged the supplementary figures to account for both our new data and the figures that were too small to read.

Reviewers' Comments:

Reviewer #1:

Remarks to the Author:

The authors adequately addressed most of the points raised, except for one (major point 2). This manuscript describes in a thorough way a strategy of replacing a mutant coding exon with a cDNA sequence and a selectable marker for therapeutic purposes (here treating RAG-SCID). Although the proposed strategy is elegant, the authors do not show functional data supporting advantages of this insertion method over their previously published insertion method or other available and functional gene therapy vectors.

To demonstrate a wider applicability the authors included data on RAG1 replacement in the revised manuscript. Because RAG1 and RAG2 are only 12 kb apart, I would not consider RAG1 another independent locus and would have preferred insertion into another locus to show broader applicability.

Reviewer #2:

Remarks to the Author:

Summary

The authors have added additional off target sequencing data showing that off target integrations appear to be similar to other previously reported, and clinically used, methods. While functional data showing that the gene replacement strategy is superior to the prior gene insertion strategy is not presented, the authors have added additional discussion of the theoretical advantages, along with demonstration of the technical generalizability of the gene replacement strategy at an additional gene locus, Rag1. I do have a continuing issue with the authors unusual way of quantifying some of their flow cytometry data in Fig 1B and Sup Figs 1-3 which should be addressed before publication, but overall the manuscript is sound and doesn't require any additional experimentation to address my remaining concerns.

Major Concerns

Major concerns have been addressed.

Minor Concerns

We're getting into the weeds here and this shouldn't significantly hold up publication of the story, but I continue to disagree with the quantifications of GFP+ cells in Figure 1B and Sup Figs 1-3. I'm sorry, but again - adjusting gates between samples is not standard practice, and these figures in the current manuscript are dangerously close to misrepresenting the underlying data.

I understand the authors want to make their rAAV6 only conditions have a "% GFP+" of ~0% so that such controls look cleaner - but because of the episomal expression, there are GFP+ cells in these conditions - drawing your gates to exclude them without clear descriptions of what you've done misrepresents to the reader the nature of these experiments. At a minimum these axes should state "% GFP+ Cells above rAAV6 Only". Drawing your positive gate based on AAV only is fine - but your reference to Cromer et al does not support the author's choice to vary where that positive gate between samples - Cromer et al does not vary their gates from one vector to another as the authors do (See ED Fig 3B from Cromer et al).

There's also a contradiction - the Sup Fig 1E legend states that the cells in the positive gates in the top panels are the episomal expression cells and thus says "Episomal expression is determined to be 1%" - but the authors argument in the response letter clearly states that you set the gate to exclude episomal expression cells. Which is it? You can't arbitrarily decide that episomal expression must equal 1% and then draw a gate that gives you that number...

As stated in the previous my review - a negative control with no GFP expression here is essential to set your gates, and the included plots in the review response based on such a negative control (which should be displayed as a flow plot in Sup Fig 1E) make all of the points the authors wish to

make while also displaying the data accurately. I would recommend including the plots from the review response letter instead of the current ones, which I fear are close to misrepresenting the underlying data. The word "episomal" also does not appear once in the paper's text despite being the core reason for making a non-obvious choice in your flow quantifications. The authors response to this reviewers earlier Minor Concern is a valid justification of their non-standard quantification choices - but none of that appears in the main text, figure legends, or methods section. The text should also be altered to make this apartment in addition to changing the figures.

This could all just be academic if all of the integration methods clearly positive populations above the level observed with episomal expression alone. But this is not the case - the CSI GFP+ population in clearly cut in half in Supplementary Figure 1E. Representing this clearly wrong quantification as the real "% GFP+ cells" in Figure 1B gives the reader an inaccurate view. This has a real impact on interpretation because the flow populations in Sup Fig 1E seem to indicate the gene insertion strategy has a higher efficiency than the gene replacement strategy, although the expression levels are decreased. This tradeoff of efficiency vs higher expression is an important consideration for evaluating the utility of the replacement method.

Reviewer #3:

Remarks to the Author:

The authors respond to the first reviews with significant additional data, including studies of the Rag1 gene to add to those of Rag2, as well as some long-range sequencing data and ITR seq to analyze integration of the AAV donor.

However, the authors have not provided any new data to support their unsubstantiated hypothesis that the coding sequencing replacement method is superior to the coding sequence insertion method (CSI). Nor have they demonstrated that the Rag2 gene inserted using their CSR strategy leads to any improved expression pattern.

It is not known if the CSI approach would affect RAG2 expression, as the cis acting elements are still nearby, although the length of the transgene further away – this distance may or may not matter. One cited reference (Miyazaki K, *Sci Immunol.* 2020;5(51)) to support the critical locus architecture for expression only addresses the role of E2A binding sites. A disadvantage of the CSR strategy is that it required a long right homology arm (1600 bp to reach the frequency of integration achieved with an 800 bp RHA for the CSI strategy. This need for a long RHA could be limiting for the carrying capacity of AAV vectors for donor delivery. Additionally, if there is concern that any abnormalities in the architecture of the RAG2 gene locus by insertion may alter expression, the inclusion of an exogenous report such a tNGFR, may cause this same issue. Thus, the new title is misleading as the paper does not actually show either that the CSR approach "Preserve Endogenous Gene Regulation and Locus Structure for Therapeutic Applications". In my opinion, the authors need to be clearer about the hypothetical nature of the putative advantage of the CDS approach and state frankly (in Abstract and Discussion) that the current work does not demonstrate any benefits yet, just evidence that this approach may be used and need further study to know if it is better the CSI.

2. We asked specifically, if they have verified by direct sequencing theRAG2 transgene inserted by CSR in seamless with perfect sequence across the transgene and into the 3' junction to the genome. They describe "NHEJ-based insertions to the cut site and partial NHEJ" at 5-10% frequencies, but do not affirm that the insertions by the intended HDR thru RHA and the 3' end of RAG2 gene downstream from the Cas9 cut site were otherwise sequence perfect or of some of the long reads revealed imprecisions in the HDR insertions. Specific data are needed here as to what these sequences showed. Were 100% of the sequence reads seamless and without any junctional abnormalities, or were there some with other types of imperfect recombinations? These data should be contained in the long-range sequencing performed, but they need to be described. And what is meant by "partial NHEJ"?

Reviewer comments point-by-point letter:

Reviewer #1 (Remarks to the Author):

The authors adequately addressed most of the points raised, except for one (major point 2). This manuscript describes in a thorough way a strategy of replacing a mutant coding exon with a cDNA sequence and a selectable marker for therapeutic purposes (here treating RAG-SCID). Although the proposed strategy is elegant, the authors do not show functional data supporting advantages of this insertion method over their previously published insertion method or other available and functional gene therapy vectors.

To demonstrate a wider applicability the authors included data on RAG1 replacement in the revised manuscript. Because RAG1 and RAG2 are only 12 kb apart, I would not consider RAG1 another independent locus and would have preferred insertion into another locus to show broader applicability.

We thank Reviewer #1 for their important feedback, however, at this stage, showing the broad applicability requested is beyond the scope of this publication and will be established in future works.

Reviewer #2 (Remarks to the Author):

Summary

The authors have added additional off target sequencing data showing that off target integrations appear to be similar to other previously reported, and clinically used, methods. While functional data showing that the gene replacement strategy is superior to the prior gene insertion strategy is not presented, the authors have added additional discussion of the theoretical advantages, along with demonstration of the technical generalizability of the gene replacement strategy at an additional gene locus, Rag1. I do have a continuing issue with the authors unusual way of quantifying some of their flow cytometry data in Fig 1B and Sup Figs 1-3 which should be addressed before publication, but

overall the the manuscript is sound and doesn't require any additional experimentation to address my remaining concerns.

We want to thank Reviewer #2 for the very encouraging feedback and to express our gratitude for taking the time to review our manuscript.

Major Concerns

Major concerns have been addressed.

Minor Concerns

We're getting into the weeds here and this shouldn't significantly hold up publication of the story, but I continue to disagree with the quantifications of GFP+ cells in Figure 1B and Sup Figs 1-3. I'm sorry, but again - adjusting gates between samples is not standard practice, and these figures in the current manuscript are dangerously close to misrepresenting the underlying data. I understand the authors want to make their rAAV6 only conditions have a "% GFP+" of ~0% so that such controls look cleaner - but because of the episomal expression, there are GFP+ cells in these conditions - drawing your gates to exclude them without clear descriptions of what you've done misrepresents to the reader the nature of these experiments. At a minimum these axes should state "% GFP+ Cells above rAAV6 Only". Drawing your positive gate based on AAV only is fine - but your reference to Cromer et al does not support the author's choice to vary where that positive gate between samples - Cromer et al does not vary their gates from one vector to another as the authors do (See ED Fig 3B from Cromer et al).

There's also a contradiction - the Sup Fig 1E legend states that the cells in the positive gates in the top panels are the episomal expression cells and thus says "Episomal expression is determined to be 1%" - but the authors argument in the response letter clearly states that you set the gate to exclude

episomal expression cells. Which is it? You can't arbitrarily decide that episomal expression must equal 1% and then draw a gate that gives you that number...

As stated in the previous my review - a negative control with no GFP expression here is essential to set your gates, and the included plots in the review response based on such a negative control (which should be displayed as a flow plot in Sup Fig 1E) make all of the points the authors wish to make while also displaying the data accurately. I would recommend including the plots from the review response letter instead of the current ones, which I fear are close to misrepresenting the underlying data. The word "episomal" also does not appear once in the paper's text despite being the core reason for making a non-obvious choice in your flow quantifications. The authors response to this reviewers earlier Minor Concern is a valid justification of their non-standard quantification choices - but none of that appears in the main text, figure legends, or methods section. The text should also be altered to make this apartment in addition to changing the figures.

This could all just be academic if all of the integration methods clearly positive populations above the level observed with episomal expression alone. But this is not the case - the CSI GFP+ population in clearly cut in half in Supplementary Figure 1E. Representing this clearly wrong quantification as the real "% GFP+ cells" in Figure 1B gives the reader an inaccurate view. This has a real impact on interpretation because the flow populations in Sup Fig 1E seem to indicate the gene insertion strategy has a higher efficiency than the gene replacement strategy, although the expression levels are decreased. This tradeoff of efficiency vs higher expression is an important consideration for evaluating the utility of the replacement method.

We thank the reviewer for these points and hope to clarify. To be as clear as possible and avoid any misinterpretations of our data, we have added Supplementary Note 1 where we provide an in-depth explanation of how we established our "unconventional" gating strategy. To make this clear throughout the manuscript, we have done the following:

1. We have changed the Y-Axis on Figs. 1B, 2B, and Supplementary Fig. 2B to "% GFP⁺ Cells Above rAAV6 Only Cells" and on Figs. 1D, 2D, and Supplementary 2D to "MFI (GFP⁺ Cells [above rAAV6 only cells])" to make clear that we are presenting GFP⁺ values of only cells above their respective rAAV6 only samples.
2. We corrected the reference to the literature that supports gating to eliminate the GFP^{low} population from Cromer *et al.* to Dever *et al.* (PMID: 27820943), Bak *et al.* (PMID: 29370156), and Bak *et al.* (PMID: 28956530) in Supplementary Note 1.
3. We rephrased the figure legends to no longer refer to the "1% episomal gating" and instead we say: "Gating determination is based on the cells treated with the rAAV6 vector alone for each donor to determine the level of episomal expression."
4. We added the following sentence to the main text: "Each rAAV6 donor was gated based on its respective rAAV only (RNP⁻) sample to eliminate cells that are only expressing the donor in an episomal manner (GFP^{low} cells) (see Supplementary Note 1 and Appendix 1 for description of gating strategy)."
5. We added a representative flow cytometry plot of a *Mock* sample in Supplementary Figs. 1 and 3 for comparative purposes to show the difference that episomal expression causes in the rAAV6 only samples.
6. We added Appendix 2 which depicts the significant differences in GFP⁺ frequencies in rAAV6 only samples when we use the untreated *Mock* samples for uniform gating. This is to again highlight our reasoning for our unconventional strategy since these cells that are believed to be HDR-negative should not be included in the MFI calculation.

Lastly, to address the reviewer's concern regarding the tradeoff between efficiency and expression, we compared the GFP⁺ frequencies and the MFI values between the two different gating strategies (uniform gating based on the untreated *Mock* sample and gating each sample based on its respective rAAV6 only sample). When comparing these strategies for the insertion donor (CSI_GFP-

BGHpA_400x400) and the replacement donor with a 1,600bp homology arm (CDSR_GFP-BGHpA_400x1600) we found that GFP⁺ frequencies increased from 21.8% to 55.4% for the insertion donor and from 25.2% to 33.3% for the replacement donor when using a uniform gate based on the *Mock* sample. Additionally, the MFI values decreased from 0.8×10^6 to 0.4×10^6 for the insertion donor and from 2.8×10^6 to 2.2×10^6 for the replacement donor when using a uniform gate based on the *Mock* sample. In short, with the uniform gate, the difference in GFP⁺ frequencies between the two donors becomes statistically significant and the difference in MFI between insertion and replacement donors increases. Taking this into account, in our view, this calculation is imprecise for two reasons. Firstly, based on the explanation given in Supplementary Note 1, we believe that including the GFP^{low} cells (many of which are truly HDR-negative based on the literature [see Dever *et al.* {PMID: 27820943}, Bak *et al.* {PMID: 29370156}, and Bak *et al.* {PMID: 28956530}]) into the GFP⁺ frequencies and/or the MFI calculation will skew the results. Secondly, when we compared the ddPCR values (the HDR integration frequencies on the allelic level), the observed values were comparable between the two donors (Fig. 1C). Thus, we found the absolute HDR integration efficiency to be broadly comparable between the insertion donor (CSI_GFP-BGHpA_400x400) and the replacement donor with a 1,600bp homology arm (CDSR_GFP-BGHpA_400x1600). Lastly, even in the case where our gating strategy is underestimating the HDR efficiencies in the flow cytometry analyses, the uniform gating strategy still gives 33.3% HDR for the CDSR_GFP-BGHpA_400x1600 replacement donor.

Reviewer #3 (Remarks to the Author):

The authors respond to the first reviews with significant additional data, including studies of the Rag1 gene to add to those of Rag2, as well as some long-range sequencing data and ITR seq to analyze integration of the AAV donor.

However, the authors have not provided any new data to support their unsubstantiated hypothesis that the coding sequencing replacement method is superior to the coding sequence insertion method (CSI). Nor have they demonstrated that the Rag2 gene inserted using their CSR strategy leads to any improved expression pattern.

It is not known if the CSI approach would affect RAG2 expression, as the cis acting elements are still nearby, although the length of the transgene further away – this distance may or may not matter. One cited reference (Miyazaki K, Sci Immunol. 2020;5(51)) to support the critical locus architecture for expression only addresses the role of E2A binding sites. A disadvantage of the CSR strategy is that it required a long right homology arm (1600 bp to reach the frequency of integration achieved with an 800 bp RHA for the CSI strategy. This need for a long RHA could be limiting for the carrying capacity of AAV vectors for donor delivery. Additionally, if there is concern that any abnormalities in the architecture of the RAG2 gene locus by insertion may alter expression, the inclusion of an exogenous report such a tNGFR, may cause this same issue. Thus, the new title is misleading as the paper does not actually show either that the CSR approach “Preserve Endogenous Gene Regulation and Locus Structure for Therapeutic Applications”. In my opinion, the authors need to be clearer about the hypothetical nature of the putative advantage of the CDS approach and state frankly (in Abstract and Discussion) that the current work does not demonstrate any benefits yet, just evidence that this approach may be used and need further study to know if it is better the CSI.

We want to thank Reviewer #3 for the feedback and to express our gratitude for taking the time to review our manuscript a second time. We have taken their excellent points into consideration and have done our best to qualify, caveat, and tone down our claims where needed for optimal clarity. Additionally, we have changed the title of our manuscript to “CRISPR-Cas9 Engineering of the RAG2 Locus via Complete Coding Sequence Replacement for Therapeutic Applications” to highlight these claims more accurately.

2. We asked specifically, if they have verified by direct sequencing the RAG2 transgene inserted by HDR in seamless with perfect sequence across the transgene and into the 3' junction to the genome. They describe "NHEJ-based insertions to the cut site and partial NHEJ" at 5-10% frequencies, but do not affirm that the insertions by the intended HDR thru RHA and the 3' end of RAG2 gene downstream from the Cas9 cut site were otherwise sequence perfect or if some of the long reads revealed imperfections in the HDR insertions. Specific data are needed here as to what these sequences showed. Were 100% of the sequence reads seamless and without any junctional abnormalities, or were there some with other types of imperfect recombinations? These data should be contained in the long-range sequencing performed, but they need to be described.

And what is meant by "partial NHEJ"?

We thank the reviewer for this comment and hope we can clarify. We sequenced using the ONT MINION; and while it is a very valuable technology, it is well established in the field of long-read sequencing that this technology has a relatively high error rate (see Delahaye *et al.* [PMID: 34597327]). Excusing these expected sequencing errors, we classified each read based on manual alignment and assessment. Reads that were classified as HDR were ~100% accurate and showed no junctional aberrations. We have included a depiction of one representative read aligned via NCBI Blastn

(https://blast.ncbi.nlm.nih.gov/Blast.cgi?PROGRAM=blastn&PAGE_TYPE=BlastSearch&LINK_LOC=blasthome) to a reference sequence of perfect HDR for both the CSI_Corr and CDSR_Corr_Endo3'UTR donors in Supplementary Fig. 5E-F to highlight this specificity (we also provided the exact read sequences in Supplementary Data 2. Additionally, for clarity's sake, we added Appendix 1 which depicts the different detected gene-editing products. As can be seen, NHEJ/HDR specifically refers to sequences that displayed NHEJ repair on one end and successful HDR on the other end. NHEJ-based insertions, on the other hand, are products that displayed either large

insertions of DNA segments (>50bp) to the cut site or had the donor, or part of the donor, integrate to the cut site via NHEJ on both ends of the fragment.

Reviewers' Comments:

Reviewer #2:

Remarks to the Author:

The additional clarifications to figures, text changes, and the added supplementary note all give the reader a more clear understanding of the reasons for setting different cutoff values/gates for different samples. It is now clear in the text, as opposed to only in the review response letter, the tradeoffs in quantifications due to high and variable levels of episomal expression when using AAV vectors. As stated previously, the authors present a valuable technical exploration of a targeted gene replacement strategy in contrast to more traditional gene insertion approaches, and hopefully future work from their group and others will show experimentally the functional implications of these differing gene therapy approaches. I have no further concerns with the manuscript, and the author's study should be well received by the broad audience of Nature Communications.

Reviewer #3:

Remarks to the Author:

The authors have responded to reviewers comments adequately. I think the modified title is more appropriate for the scope of data.

Reviewer comments point-by-point letter:

Reviewer #2 (Remarks to the Author):

The additional clarifications to figures, text changes, and the added supplementary note all give the reader a more clear understanding of the reasons for setting different cutoff values/gates for different samples. It is now clear in the text, as opposed to only in the review response letter, the tradeoffs in quantifications due to high and variable levels of episomal expression when using AAV vectors. As stated previously, the authors present a valuable technical exploration of a targeted gene replacement strategy in contrast to more traditional gene insertion approaches, and hopefully future work from their group and others will show experimentally the functional implications of these differing gene therapy approaches. I have no further concerns with the manuscript, and the author's study should be well received by the broad audience of Nature Communications.

We want to thank reviewer #2 for their hard work reviewing our manuscript and we very much appreciate their endorsement for publication.

Reviewer #3 (Remarks to the Author):

The authors have responded to reviewers comments adequately.

I think the modified title is more appropriate for the scope of data.

We want to thank reviewer #3 for their extensive work reviewing and commenting on our manuscript.